# Autonomous Learning of Object-Centric Abstractions for High-Level Planning

**Steven James & Benjamin Rosman**
University of the Witwatersrand
Johannesburg, South Africa
{steven.james,benjamin.rosman1}@wits.ac.za

**George Konidaris**
Brown University
Providence RI 02912, USA
gdk@cs.brown.edu

## Abstract

We propose a method for autonomously learning an object-centric representation of a continuous and high-dimensional environment that is suitable for planning. Such representations can immediately be transferred between tasks that share the same types of objects, resulting in agents that require fewer samples to learn a model of a new task. We first demonstrate our approach on a 2D crafting domain consisting of numerous objects where the agent learns a compact, lifted representation that generalises across objects. We then apply it to a series of Minecraft tasks to learn object-centric representations and object types—directly from pixel data—that can be leveraged to solve new tasks quickly. The resulting learned representations enable the use of a task-level planner, resulting in an agent capable of transferring learned representations to form complex, long-term plans.

## 1 Introduction

Model-based methods are a promising approach to improving sample efficiency in reinforcement learning (RL). However, they require the agent to either learn a highly detailed model—which is infeasible for sufficiently complex problems (Ho et al., 2019)—or to build a compact, high-level model that abstracts away unimportant details while retaining only the information required to plan. This raises the question of how best to build such an abstract model. While recent advances have shown how to learn models of complex environments, they lack theoretical guarantees and require millions of sample interactions (Schrittwieser et al., 2020; Hafner et al., 2021). Fortunately, recent work has shown how to learn an abstraction of a task that is provably suitable for planning with a given set of skills (Konidaris et al., 2018). However, these representations are highly task-specific and must be relearned for any new task, or even any small change to an existing task. This makes them fatally impractical, especially for agents that must solve multiple complex tasks.

We extend these methods by incorporating additional structure—namely, that the world consists of objects, and that similar objects are common amongst tasks. For example, when we play video games we solve the game quickly by leveraging our existing knowledge of objects and their affordances, such as doors and ladders which occur across multiple levels (Dubey et al., 2018). Similarly, robot manipulation tasks often use the same robot and a similar set of physical objects in different configurations. This can substantially improve learning efficiency, because an object-centric model can be reused wherever that same object appears (within the same task, or across different tasks) and can also be generalised across objects that behave similarly—object *types*.

We assume that the agent is able to individuate the objects in its environment, and propose a framework for building portable object-centric abstractions given only the data collected by executing high-level skills. These abstractions specify both the abstract object attributes that support high-level planning, and an object-relative lifted transition model that can be instantiated in a new task. This reduces the number of samples required to learn a new task by allowing the agent to avoid relearning the dynamics of previously seen object types. We make the following contributions: under the assumption that the agent can individuate objects in its environment, we develop a framework for building portable, object-centric abstractions, and for estimating object types, given only the data collected by executing high-level skills. We also show how to integrate problem-specific information to instantiate these

representations in a new task. This reduces the samples required to learn a new task by allowing the agent to avoid relearning the dynamics of previously seen objects.

We demonstrate our approach on a Blocks World domain and a 2D crafting domain, and then apply it to a series of Minecraft tasks where an agent autonomously learns an abstract representation of a high-dimensional task from raw pixel input. In particular, we use the probabilistic planning domain definition language (PPDDL) (Younes & Littman, 2004) to represent our learned abstraction, which allows for the use of existing task-level planners. Our results show that an agent can leverage these portable abstractions to learn a representation of new Minecraft tasks using a diminishing number of samples, allowing it to quickly construct plans composed of hundreds of low-level actions.[1]

## 2 BACKGROUND

We assume that tasks are modelled as semi-Markov decision processes $\mathcal{M} = \langle \mathcal{S}, \mathcal{O}, \mathcal{T}, \mathcal{R} \rangle$ where (i) $\mathcal{S}$ is the state space; (ii) $\mathcal{O}(s)$ is the set of temporally extended actions known as *options* available at state $s$; (iii) $\mathcal{T}$ describes the transition dynamics, specifying the probability of arriving in state $s'$ after option $o$ is executed from $s$; and (iv) $\mathcal{R}$ specifies the reward for reaching state $s'$ after executing option $o$ in state $s$. An option $o$ is defined by the tuple $\langle I_o, \pi_o; \beta_o \rangle$, where $I_o$ is the *initiation set* specifying the states where the option can be executed, $\pi_o$ is the *option policy* which specifies the action to execute, and $\beta_o$ the probability of the option terminating in each state (Sutton et al., 1999).

We adopt the object-centric formulation from Ugur & Piater (2015): in a task with $n$ objects, the state is represented by the set $\{\mathbf{f}_a, \mathbf{f}_1, \mathbf{f}_2, \ldots, \mathbf{f}_n\}$, where $\mathbf{f}_a$ is a vector of the agent's features and $\mathbf{f}_i$ is a vector of features particular to object $i$. Note that the feature vector describing each object can itself be arbitrarily complex, such as an image or voxel grid—in one of our domains we use pixels.

Our state space representation assumes that individual objects have already been factored into their constituent low-level attributes. Practically, this means that the agent is aware that the world consists of objects, but is unaware of what the objects are, or whether multiple instantiations of the same object are present. It is also easy to see that different tasks will have differing numbers of objects with potentially arbitrary ordering; any learned abstract representation should be agnostic to this.

### 2.1 STATE ABSTRACTIONS FOR PLANNING

We intend to learn an abstract representation suitable for planning. Prior work has shown that a sound and complete abstract representation must necessarily be able to estimate the set of initiating and terminating states for each option (Konidaris et al., 2018). In classical planning, this corresponds to the *precondition* and *effect* of each high-level action operator (McDermott et al., 1998).

The precondition is defined as $\text{Pre}(o) = \Pr(s \in I_o)$, which is a probabilistic classifier that expresses the probability that option $o$ can be executed at state $s$. Similarly, the effect or *image* represents the distribution of states an agent may find itself in after executing an option from states drawn from some starting distribution (Konidaris et al., 2018). Since the precondition is a probabilistic classifier and the effect is a density estimator, they can be learned directly from option execution data.

We can use preconditions and effects to evaluate the probability of a sequence of options—a plan— executing successfully. Given an initial state distribution, the precondition is used to evaluate the probability that the first option can execute, and the effects are used to determine the resulting state distribution. We can apply the same logic to the subsequent options to compute the probability of the entire plan executing successfully. It follows that these representations are sufficient for evaluating the probability of successfully executing *any* plan (Konidaris et al., 2018).

**Partitioned Options** For large or continuous state spaces, estimating $\Pr(s' \mid s, o)$ is difficult because the worst case requires learning a distribution conditioned on every state. However, if we assume that terminating states are independent of starting states, we can make the simplification $\Pr(s' \mid s, o) = \Pr(s' \mid o)$. These *subgoal* options (Precup, 2000) are not overly restrictive, since they refer to options that drive an agent to some set of states with high reliability. Nonetheless, many options are not subgoal. It is often possible, however, to *partition* an option's initiation set

---

[1] More results and videos can be found at: `https://sites.google.com/view/mine-pddl`

into a finite number of subsets, so that it is approximately subgoal when executed from any of the individual subsets. That is, we partition an option $o$'s start states into finite regions $\mathcal{C}$ such that $\Pr(s' \mid s, o, c) \approx \Pr(s' \mid o, c), c \in \mathcal{C}$ (Konidaris et al., 2018).

**Factors**   We adopt the frame assumption—an assumption implicit in classical planning—which states that aspects of the world not explicitly affected by an agent's action remain the same (Pasula et al., 2004). Prior work leverages this to learn a factored or STRIPS-like (Fikes & Nilsson, 1971) representation by computing the option's *mask*: the state variables explicitly changed by the option (Konidaris et al., 2018). In our formulation, the state space is already factorised into its constituent objects, so computing the mask amounts to determining which objects are affected by a given option.

## 3   LEARNING OBJECT-CENTRIC REPRESENTATIONS

Although prior work (Konidaris et al., 2018) allows an agent to autonomously learn an abstract representation supporting fast task-level planning, that representation lacks generalisability—since the symbols are distributions over states in the current task, they cannot be reused in new ones. This approach can be fatally expensive in complex domains, where learning an abstract model may be as hard as solving a task from scratch, and is therefore pointless if we only want to solve a single task. However, an agent able to reuse aspects of its learned representation can amortise the cost of learning over many interactions, accelerating learning in later tasks. The key question is what forms of representation support transfer in this way.

We now introduce an object-centric generalisation of a learned symbolic representation that admits transfer in tasks when the state space representation consists of features centred on objects in the environment. This is common in robotics, where each object is often isolated from the environment and represented as a point cloud or subsequently a voxelised occupancy grid. Our approach builds on a significant amount of machinery, involving clustering, feature selection, classification and density estimation. We summarise our proposed approach in Figure 1 and provide a high-level description in the remainder of this section, but provide pseudocode and specific practical details in the appendix.

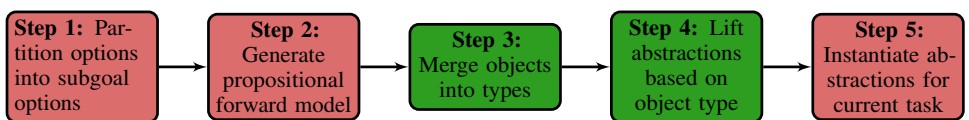

Figure 1: Learning lifted representations from data. Red nodes represent problem-specific representations, while green nodes are abstractions that can be transferred between tasks.

### 3.1   GENERATING A PROPOSITIONAL MODEL (STEPS 1–2) (AS IN KONIDARIS ET AL., 2018)

The agent begins by collecting transition data using an exploration policy. The first step is to partition the options into approximately subgoal options. For each option $o$ and empirical sets of initial and terminating states $\tilde{I}_o$ and $\tilde{\beta}_o$, the agent partitions $\tilde{I}_o$ into a number of disjoint subsets, such that for each subset $K \subseteq \tilde{I}_o$, we have $\Pr(s' \mid s_i, o) = \Pr(s' \mid s_j, o) \forall s_i, s_j \in K, s' \in \tilde{\beta}_o$. In other words, the effect distribution of the option is identical, independent of the state in $K$ from which it was executed. In practice, this can be approximated by first clustering state transition samples based on terminating states, and then assigning each cluster to a partition. Finally, pairs of partitions whose initiating states overlap are merged to handle probabilistic effects (Konidaris et al., 2018).

The agent next learns a precondition classifier for each approximately partitioned option. For each partition, the initiating states are used as positive examples, and all other states are treated as negative ones. A feature selection procedure next determines which objects are relevant to the precondition, and a classifier is fit using only those objects. A density estimator is used to estimate the effect distribution for each partitioned option; the agent learns distributions over only the objects affected by the option. Together these effect distributions form our propositional PPDDL vocabulary $\mathcal{V}$.

To construct a PPDDL representation for each partitioned option, we must specify both the precondition and effects using the state distributions (propositions) in $\mathcal{V}$. The effects are directly specified

using these distributions, and so pose no problem. However, the estimated precondition is a classifier rather than a state distribution. The agent must therefore iterate through all possible effect distributions to compute whether the skill can be executed there. To do so, we denote $\mathcal{P}$ as some set of propositions in $\mathcal{V}$, and $\mathcal{G}(s; \mathcal{P})$ as the probability that a low-level state $s$ is drawn from the conjunction of propositions in $\mathcal{P}$. Then, for an option with learned classifier $\hat{I}_o$, we can represent the precondition with every $\mathcal{P} \in \wp(\mathcal{V})$ such that $\int_{\mathcal{S}} \hat{I}_o(s) \mathcal{G}(s; \mathcal{P}) ds > 0$, where $\wp(\mathcal{V})$ denotes the powerset of $\mathcal{V}$. In other words, the agent considers every combination of effect state distributions and draws samples from their conjunction. If these samples are classified as positive by $\hat{I}_o$, then the conjunction of $\mathcal{P}$ is used to represent the precondition. The preconditions and effects are now specified using distributions over state variables, where each distribution is a proposition—this is our PPDDL representation, which is sound and complete for planning.

## 3.2 GENERATING A LIFTED, TYPED MODEL (STEPS 3–4)

At this point the agent has learned an abstract, but task-specific, representation. Unfortunately there is no opportunity for transfer (both within the task and between different tasks), because each object is treated as unique. To overcome this, we now propose a method for estimating object *types* using the PPDDL preconditions and effects learned in the previous section.

**Definition 1.** Assume that option $o$ has been partitioned into $n$ subgoal options $o(1), \ldots, o(n)$. Object $i$'s *profile* under option $o$ is denoted by the set

$$\text{Profile}(i, o) = \left\{ \{\text{Pre}_i^{o(1)}, \mathcal{E}_i^{o(1)}\}, \ldots, \{\text{Pre}_i^{o(n)}, \mathcal{E}_i^{o(n)}\} \right\},$$

where $\text{Pre}_i^{o(k)}$ is the distribution over object $i$'s states present in the precondition for partition $k$, and $\mathcal{E}_i^{o(k)}$ is object $i$'s effect distribution. Note that these preconditions and effects can be null.

**Definition 2.** Two objects $i$ and $j$ are *option-equivalent* if, for a given option $o$, $\text{Profile}(i, o) = \text{Profile}(j, o)$. Furthermore, two objects are *equivalent* if they are option-equivalent for every $o$ in $\mathcal{O}$.

The above definition implies that objects are equivalent if one object can be substituted for another while preserving every operator's abstract preconditions and effects. Such objects can be grouped into the same *object type*, since they are functionally indistinguishable for the purposes of planning. In practice, however, we can use a weaker condition to construct object types. Since an object-centric skill will usually modify only the object being acted upon, and because we have subgoal options, we can take a similar approach to Ugur & Piater (2015) and group objects by effects only:

**Definition 3.** Assume that option $o$ has been partitioned into $n$ subgoal options $o(1), \ldots, o(n)$. Object $i$'s *effect profile* under option $o$ is denoted by the set

$$\text{EffectProfile}(i, o) = \left\{ \mathcal{E}_i^{o(1)}, \ldots, \mathcal{E}_i^{o(n)} \right\},$$

where $\mathcal{E}_i^{o(k)}$ is object $i$'s effect distribution. Two objects $i$ and $j$ are *effect-equivalent* if $\text{EffectProfile}(i, o) = \text{EffectProfile}(j, o)$ for every $o$ in $\mathcal{O}$.

The notion of effect equivalence was first proposed by Şahin et al. (2007). Such an approach assumes that an object's type depends on both the intrinsic properties of the object, as well as the agent's endowed behaviours (Chemero, 2003). However, this definition does not account for preconditions, nor does it consider interactions involving multiple objects. The latter issue is also present in frameworks such as object-oriented MDPs(Diuk et al., 2008), where the dynamics are described by pairwise object interactions (one of those objects being the agent). Nonetheless, the approach will prove sufficient for our purposes, and we leave a more complete definition to future work.

By computing effect profiles using the propositional representation, the agent can determine whether objects $i$ and $j$ are similar (using an appropriate measure of distribution similarity) and, if so, merge them into the same object type. Propositions representing distributions over individual objects can now be replaced with predicates that are parameterised by types. For example, consider a domain with three objects—two identical doors and a block—and an agent with a single option to open a door. Since the option can affect both of the doors, it would first be partitioned into two subgoal options (one for each door). Given this, the effect profile for the first and second doors would be $\{\texttt{open1}, \varnothing\}$ and $\{\varnothing, \texttt{open2}\}$ respectively. The effect profile for the block, which cannot be acted upon, would

simply be $\{\varnothing, \varnothing\}$. By noting that the sets representing the doors' effects are equal, the agent can merge the two door objects into a single type and replace the `open1` and `open2` propositions with a single `open` predicate parameterised by objects of type `door`.

### 3.3 PROBLEM-SPECIFIC INSTANTIATION (STEP 5)

If the task dynamics are completely described by the state of each object, as is the case in object-oriented MDPs (Diuk et al., 2008), then our typed representation is sufficient for planning. However, in many domains the object-centric state space is *not* Markov. For example, in a task where only a particular key opens a particular door, the state of the objects alone is insufficient to describe dynamics—the identities of the key and door are necessary too. A common strategy in this case is to augment an ego- or object-centric state space with problem-specific, allocentric information to preserve the Markov property (Konidaris et al., 2012; James et al., 2020). We denote $\mathcal{X}$ as the space of problem-specific state variables. $\mathcal{S}$ remains the original object-centric state space. The above complication does not negate the benefit of learning transferable abstract representations, as our existing operators learned in $\mathcal{S}$ can be augmented with propositions over $\mathcal{X}$ on a per-task basis. In general, local information relative to individual objects will transfer between tasks, but problem-specific information, such as an object's global location, must be relearned each time.

For a given partioned option, the agent repeats the partitioning procedure, but this time using only problem-specific state data. This forms $n$ partitioned options that are subgoal in both $\mathcal{S}$ *and* $\mathcal{X}$. Denote $\kappa_i$ and $\lambda_i$ for $i \in \{1, \ldots, n\}$ as the sets of start and end states for each of these newly partitioned options. The agent can now *ground* the operator by appending each $\kappa_i$ and $\lambda_i$ to the precondition and effect, treating each $\kappa_i$ and $\lambda_i$ as problem-specific propositions. Finally, these problem-specific propositions must be linked with the grounded objects being acted upon. The agent therefore adds a precondition predicate conditioned on the identity of the grounded objects, where objects' identities are simply their respective index in the state vector.

## 4 EXPERIMENTS

We first demonstrate our framework on the classic Blocks World domain (Section 4.1). While the high-level operators and predicates describing the domain are usually given, we show how such a representation can be learned autonomously from scratch. In Section 4.2, we apply our method to a 2D crafting environment (Andreas et al., 2017) consisting of over 20 objects, where an agent learns a representation that generalises across objects and can be used to solve hierarchical tasks. We then demonstrate that our method scales to significantly harder problems by applying it to a high-dimensional Minecraft task (Section 4.3). Finally, we investigate the transferability of the learned abstractions by transferring them to additional procedurally generated Minecraft tasks (Section 4.4). Owing to space constraints, we defer the exact implementation and domain details to the appendix.

### 4.1 LEARNING A REPRESENTATION OF BLOCKS WORLD

Blocks World consists of several blocks which can be stacked on top of one another by an agent (hand). The agent possesses options that allow it to pick up a block (`Pick`), put a block back on the table (`Put`), and stack one block on another (`Stack`). Blocks cannot be picked up if they are covered or if the hand is occupied, and can only be put down or stacked if already gripped. We consider the task consisting of three blocks `A`, `B` and `C`, where each block is described by whether there is nothing, another block, or a table directly above or below it. This allows us to determine whether a given block is on a table, on another block, or in the hand, and similarly whether another block has been stacked upon it. The hand is characterised by a Boolean indicating whether it is holding a block. Thus a state is described by $\{\mathbf{f}_H, \mathbf{f}_A, \mathbf{f}_B, \mathbf{f}_C\}$, corresponding to the hand and blocks' features respectively.

**Generating a Propositional Model (Steps 1–2)** Using the approach outlined in Section 3.1, the agent partitions the options using transition data collected from the environment. This results in a total of 30 partitioned options. It then fits a classifier to each partition's initiation states, and a density estimator to its terminating states. Finally, the agent generates a propositional PDDL using these learned preconditions and effects. The full PDDL (learned entirely from data), as well as an illustration of a learned propositional operator, is provided in the appendix.

**Generating a Lifted Typed Model (Steps 3–4)** Using the effects from the propositional representation, the agent determines that objects A, B and C all possess the same effect profiles for all options and so can be grouped into a single type, while the hand belongs to its own type. The agent can now lift its representation by replacing the learned propositions with predicates parameterised by the above types. For example, after generating the model, there are three propositions: AOnTable, BOnTable, and COnTable. Since these are distributions over objects determined to be the same type, the agent can replace them all with a single predicate OnTable(X), which accepts block objects. As a result, the agent can reduce the number of operators from 30 to 6, resulting in a more compact representation with a smaller branching factor. Figure 2 illustrates a lifted operator for picking any block X off any block Y, while the full parameterised PDDL is listed in the appendix.

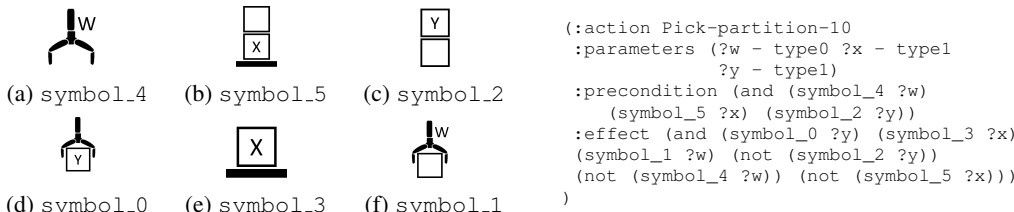

Figure 2: The learned lifted operator for a Pick action describing picking a block off another. In order to pick up block Y, it must be on block X which itself is on the table, and the hand must be empty. As a result, the hand is not empty, Y is now in the hand, and X is on the table and clear. type0 refers to the "hand" type, while type1 refers to the "block" type.

## 4.2 Learning a Representation of a Crafting Task

To demonstrate the applicability of our approach to tasks with a large number of objects, we extend the domain proposed by Andreas et al. (2017), where an agent is required to collect objects and craft them to create new ones. The domain consists of a $12 \times 12$ grid containing multiple objects, such as wood, grass, iron, rock, water and gold, as well as three workshops. Other items such as bridges can be crafted using collected objects, but only if the agent is standing near the appropriate workshop. The state of each object is characterised by whether it is on the grid (present), has been picked up by the agent (picked) or does not yet exist (non-existent); the state of the world is given by the state of each object (of which there are 23), as well as the agent's egocentric view. The agent is given four skills: WalkTo, Pickup, Place and Craft—an object can only be picked up when the agent is adjacent to it, and any particular item can only be crafted if the agent has picked up the correct "ingredients". Furthermore, the gold object is separated by water and can only be accessed once the agent has placed a bridge, which must first be crafted. To solve the task, the agent is required to construct a gold arrow, which involves collecting wood and iron to create a bridge, then collecting the gold, crafting a stick out of wood, and then finally crafting the arrow using the stick and gold. The optimal plan consists of 15 option executions, a portion of which is illustrated by Figure 3a.

**Generating a Propositional Model (Steps 1–2)** The agent executes options uniformly at random to collect 20 episodes' worth of transition data. The partitioning procedure results in 20 WalkTo options, 17 Pickup options, 1 Place option and 3 CraftOptions. For each partitioned option, the agent estimates the preconditions and effects, which are then used to construct a propositional PDDL consisting of 192 action operators.

**Generating a Lifted, Typed Model (Steps 3–4)** Using the effects from the propositional representation, the agent next groups objects into types based on their effect profiles. Since our approach to estimating types depends on the agent's actions, we do not recover the ground truth types such as grass and iron. Instead we discover six types: type0 represents the agent, while type1 represents objects that can be picked up and used in crafting other objects. type2 represents objects which can be picked up but are not used in crafting. This occurs only because the agent does not observe particular objects being used in this manner during the 20 episodes. The three workshops that cannot be manipulated form type3, while type4 are those objects that can be crafted. Finally, type5 represents the bridge, which is the only object that can be placed.

**Problem-Specific Instantiation (Step 5)** Since the learned representation is object-centric, it can be reused for different configurations of the domain, with varying numbers of objects. Unlike the Blocks World domain, however, the identities of the objects here are important. For example, the bridge can only be crafted at `workshop2`, and so the workshops cannot be treated interchangeably. To overcome this, the agent is required to ground the learned operators in the current task using the object identities (which can be determined using the option masks). An example of a learned, grounded operator is given by Figure 3. For any new task, the agent need only determine the identity and type of each object, which can then be combined with the learned operator to create a PDDL representation suitable for planning. Finally, we apply the miniGPT planner (Bonet & Geffner, 2005) which uses the learned representation to discover an optimal plan. More results, types and associated objects, and visualised operators are provided in the appendix.

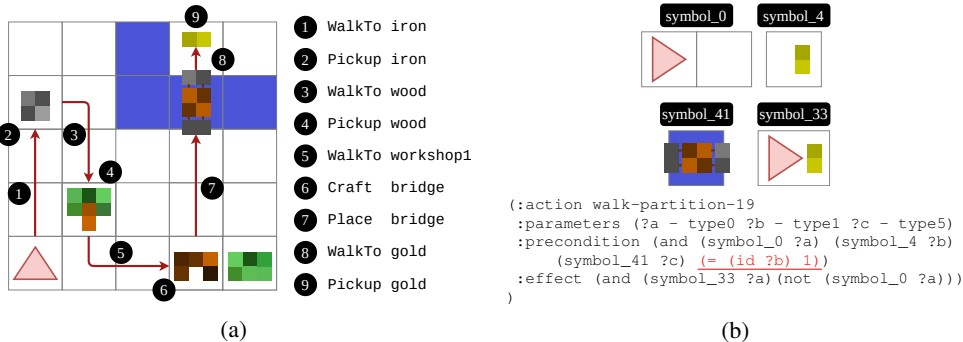

Figure 3: (a) Simplified illustration of the domain. Here, the agent executes a series of options that involves crafting a bridge to allow it to collect the gold. The full domain consists of 23 objects (excluding the agent). (b) An operator for walking to the gold. To execute, the agent must be standing in front of empty space (`symbol_0`), the gold must be present (`symbol_4`) and the bridge placed (`symbol_41`). As a result, the agent finds itself in front of the gold (`symbol_33`). The red symbol is the only predicate that must be relearned in any new configuration.

### 4.3 Learning a Representation of a Minecraft Task

In the above examples, objects are represented using pre-specified features; however, our approach is capable of scaling beyond this simple case and learning these features from pixels. We now demonstrate this in a complex Minecraft task (Johnson et al., 2016) consisting of five rooms with various items positioned throughout. Rooms are connected with either regular doors which can be opened by direct interaction, or puzzle doors requiring the agent to pull a lever to open. The world is described by the state of each of the objects (given directly by each object's appearance as a $600 \times 800$ RGB image), the agent's view, and current inventory. To simplify learning, we downscale images and applying PCA (Pearson, 1901) to a greyscaled version, preserving the top 40 principal components.

The agent is given high-level skills, such as `ToggleDoor` and `WalkToItem`. Execution is stochastic—opening doors occasionally fails, and the navigation skills are noisy in their execution. To solve the task, an agent must first collect the pickaxe, use it to break the gold and redstone blocks, and collect the resulting items. It must then navigate to the crafting table, where it uses the collected items to first craft gold ingots and subsequently a clock. Finally, it must navigate to the chest and open it to complete the task. This requires a long-horizon, hierarchical plan—the shortest plan that solves the task consists of 28 options constituting *hundreds* of low-level continuous actions.

**Generating a Propositional Model (Steps 1–2)** The agent first learns a model using the method outlined in Section 3.1. The agent partitions options using DBSCAN (Ester et al., 1996) to cluster option data based on terminating states. For each partitioned option, it then fits an SVM (Cortes & Vapnik, 1995) with Platt scaling (Platt, 1999) to estimate the preconditions, and a kernel density estimator (Rosenblatt, 1956) for effects, which are then used to construct the propositional PPDDL.

**Generating a Lifted, Typed Model (Steps 3–4)** Using the effects from the propositional representation, the agent next groups objects into types based on their effect profiles. This is made easier

because certain objects are not affected by all options. For example, the chest cannot be toggled, while a door can, and thus it is immediately clear that they are not of the same type. Having determined the types, the agent replaces all similar propositions (where similarity is measured using the KL-divergence) with a single predicate parameterised by an object of that type.

**Problem-Specific Instantiation (Step 5)**    The agent now has a representation whose operators can be transferred between tasks. However, a complication arises because the object-centric state space is *not* Markov. For example, a state where all the doors are closed and the agent is in front of the first door is indistinguishable from a state where the agent is in front of the second door. As described in Section 3.3, the agent must ground the representations in the current task by incorporating additional problem-specific state variables to preserve the Markov property. These state variables are fixed across the family of MDPs; in this case, they are the agent's $xyz$-location.

For each partitioned option, the agent again uses DBSCAN to cluster end states $\mathcal{X}$ to form partitioned subgoal options in both $\mathcal{S}$ and $\mathcal{X}$. Each of these clusters in $\mathcal{X}$ is a problem-specific proposition, which can be added to the learned operators to ground the problem. Figure 4 illustrates a learned operator for opening a particular door, where the problem-specific symbol has been tied to the door being opened in this manner. Without modifying the operator's parameter, it would be possible to open *any* door at that location. The final plan discovered by the agent is illustrated by Figure 14 in Appendix K.

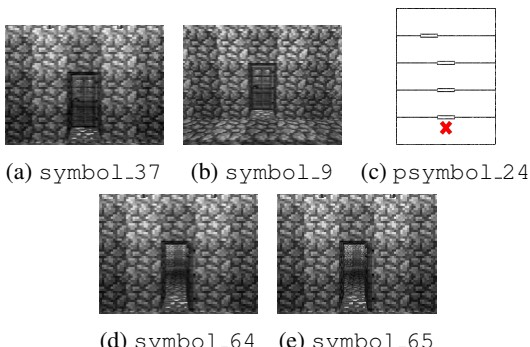

(a) `symbol_37`    (b) `symbol_9`    (c) `psymbol_24`

(d) `symbol_64`    (e) `symbol_65`

```
(:action Toggle-Door-partition-1a
:parameters (?w - type0 ?x - type1)
:precondition (and (notfailed)
        (symbol_37 ?w) (symbol_9 ?x)
        (= (id ?x) 1) (psymbol_24))
:effect (and (symbol_64 ?x)
    (symbol_65 ?w) (not (symbol_9 ?x))
    (not (symbol_37 ?w)))
)
```

(f) A learned typed PDDL operator for one partition of the `Toggle-Door` option. The predicates underlined in red must be re-learned for each new task, while the rest of the operator can be safely transferred.

Figure 4: To open a particular door, the agent must be standing in front of a closed door (`symbol_37`) at a particular location (`psymbol_24`), and the door must be closed (`symbol_9`). The effect of the skill is that the agent finds itself in front of an open door (`symbol_64`) and the door is open (`symbol_65`). `type0` and `type1` refer to the "agent" and "door" classes, while `id` is a fluent specifying the identity of the grounded door object, and is linked to the problem-specific symbol underlined in red.

### 4.4    Inter-task Transfer in Minecraft

We next investigate transferring operators between five procedurally generated tasks, where each task differs in the location of the objects and doors; the agent cannot thus simply use a plan found in one task to solve another. For a given task, the agent transfers all operators learned from previous tasks, and continues to collect samples using uniform random exploration until it produces a model which predicts that the optimal plan can be executed. Figure 5 shows the number of operators transferred between tasks, and the number of samples required to learn a model of a new task.

The minimum number of samples required to learn a model for a new task is bounded by the exploration strategy, since we must discover all problem-specific symbols to complete the model. Figure 5b shows that the number of samples required to learn a model decreases over time towards this lower bound. Inter-task transfer could be further improved by leveraging the agent's existing knowledge to perform non-uniform exploration, but we leave this to future work.

## 5    Related Work

Model-based RL methods have previously been applied to pixel-based domains. For example, Kaiser et al. (2020) and Hafner et al. (2021) learn a forward model in a latent space, while Eysenbach et al.

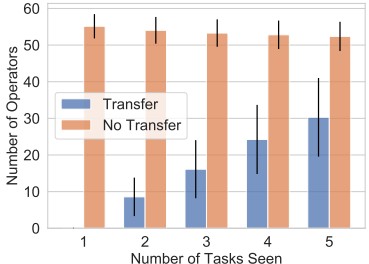

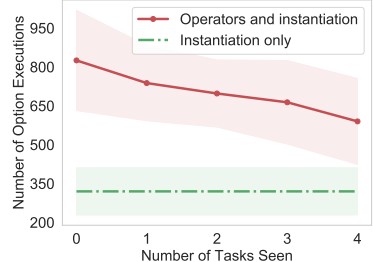

(a) Orange bars represent the number of opera-tors that must be learned by the framework of Konidaris et al. (2018) to produce a sufficiently ac-curate model to solve the task. Blue bars represent the number of operators transferred between tasks. As the number of tasks increases, the number of new operators that must be learned decreases.

(b) Number of samples required to learn suffi-ciently accurate models as a function of the number of tasks encountered. The red line represents the number of samples required to learn all the oper-ators and the instantiation, while the green line accounts for the instantiation phase only.

Figure 5: Results of learning and transferring high-level abstractions between tasks. We report the mean and standard deviation averaged over 80 runs with random task orderings.

(2019) computes a planning graph over the buffer of observed states, and Srinivas et al. (2018) learn a goal-directed latent space in which planning can occur. However, all of these approaches require hundreds of thousands of samples, which cannot feasibly be collected using the Malmo platform. By contrast, we are able to construct an abstract model using very little data.

Our method is an extension of the framework proposed by Konidaris et al. (2018), which does not allow for the learned representations to be transferred between tasks. James et al. (2020) learn an agent-centric PPDDL representation which can be transferred between tasks, but the operators have no notion of objects or types, limiting generalisability. As such, the method scales poorly with an increase in the number of objects, since their representations must be learned from scratch. In contrast, by assuming the existence of objects, we are able to learn and transfer representations across tasks that share similar types of objects.

There has been work autonomously learning parameterised, transferable representations of skills from raw data. Ugur & Piater (2015) learn object-centric PDDL representations for robotic object manipulation tasks. Similarly to our work, they estimate object types by clustering objects based on how actions affect their states, but the object features are specified prior to learning, and discrete relations between object properties are given. Furthermore, certain predicates are manually inserted to generate a sound representation. Asai (2019) learns object-centric abstractions directly from pixels, but the representations are encoded implicitly and cannot be transformed into a language that can be used by existing planners. This limitation is removed by Asai & Muise (2020), who propose an approach to extract PDDL representations from image-based transitions. However, their framework does not provide soundness guarantees, nor does it consider stochastic dynamics. By contrast, we are able to learn object-centric representations, along with the object types and the abstract high-level dynamics model, directly from raw data in a form that can be used by off-the-shelf planners.

## 6 CONCLUSION

We have introduced a method for learning high-level, object-centric representations that can be used by task-level planners. In particular, we have demonstrated how to learn the type system, predicates, and high-level operators all from pixel data. Our representation generalises across objects and can be transferred to new tasks. Although we have injected structure by assuming the existence of objects, this reflects the nature of many environments: fields such as computer vision assume that the world consists of objects, while there is evidence to suggest that infants do the same (Spelke, 1990). This allows us to convert complex, high-dimensional environments to abstract representations that serve as input to task-level planners. Our approach provides an avenue for solving sparse-reward, long-term planning problems—such as the MineRL competition (Guss et al., 2019)— that are beyond the reach of current approaches.

## ACKNOWLEDGEMENTS

This research was supported by the National Research Foundation (NRF) of South Africa under grant number 117808, and by NSF CAREER Award 1844960, DARPA under agreement numbers W911NF1820268 and (YFA) D15AP00104, AFOSR YIP agreement number FA9550-17-1-0124, and the NSF under agreement number 1955361. The U.S. Government is authorised to reproduce and distribute reprints for Governmental purposes notwithstanding any copyright notation thereon. The content is solely the responsibility of the authors and does not necessarily represent the official views of the NRF, NSF, DARPA, or the AFOSR.

The authors acknowledge the Centre for High Performance Computing (CHPC), South Africa, for providing computational resources to this research project. Computations were also performed using High Performance Computing Infrastructure provided by the Mathematical Sciences Support unit at the University of the Witwatersrand.

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

## A    ENUMERATING SUBGOAL OPTIONS FOR THE BLOCKS WORLD DOMAIN

Given the description of the Blocks World domain in the main text, we must partition the given options (`Pick`, `Put` and `Stack`) so that they adhere to the subgoal condition. When there are three blocks in the environment, we see that there are 30 partitioned options, which are described in the table below:

| Option | # partitions | Description of start states | Description of end states |
|---|---|---|---|
| `PickOffTable(X)` | 3 | `X` is on the table, `X` is clear, and the hand is empty. | `X` is grasped in the hand. |
| `PickOffSingleBlock(X, Y)` | 6 | `X` is on block `Y` which is on the table, `X` is clear, and the hand is empty. | `X` is grasped in the hand and `Y` is clear and on the table. |
| `PickOffDoubleBlock(X, Y)` | 6 | `X` is on block `Y` which is on another block, `X` is clear, and the hand is empty. | `X` is grasped in the hand and `Y` is clear and on another block. |
| `StackOnSingleBlock(X, Y)` | 6 | `X` is in the hand, and `Y` is clear and on the table. | `X` is on block `Y` which is on the table, and the hand is empty. |
| `StackOnDoubleBlock(X, Y)` | 6 | `X` is in the hand, and `Y` is clear and on another block. | `X` is on block `Y` which is on another block, and the hand is empty. |
| `Put(X)` | 3 | `X` is grasped in the hand. | `X` is on the table and the hand is empty. |

Table 1: Descriptions of the different option partitions. The description of start and end states includes only the relevant information.

## B    PROPOSITIONAL PDDL DESCRIPTION FOR THE BLOCKS WORLD TASK

Below is the automatically generated propositional PDDL description of the Blocks World domain with 3 blocks. In practice, the agent generates this description with arbitrary names for the propositions, but for readability purposes we have manually renamed them to match their semantics.

```
(define (domain BlocksWorld)
        (:requirements :strips)
        (:predicates
                (notfailed)
                (AInHand)
                (HandFull)
                (COnBlock)
                (BInHand)
                (COnTable)
                (AOnTable)
                (BOnBlock)
                (AOnBlock)
                (BOnTable)
                (CInHand)
                (HandEmpty)
                (BOnTable_BCovered)
                (COnBlock_CCovered)
                (AOnBlock_ACovered)
                (BOnBlock_BCovered)
                (COnTable_CCovered)
                (AOnTable_ACovered)
        )
```

```
(:action Pick_0
 :parameters()
 :precondition (and (HandEmpty) (AOnTable) (notfailed))
 :effect (and (AInHand) (HandFull) (not AOnTable) (not HandEmpty) (not AOnTable)
         (not HandEmpty))
)

(:action Pick_1
 :parameters()
 :precondition (and (HandEmpty) (AOnBlock) (COnBlock_CCovered) (notfailed))
 :effect (and (COnBlock) (AInHand) (HandFull) (not AOnBlock) (not HandEmpty)
         (not COnBlock_CCovered) (not AOnBlock) (not HandEmpty)
         (not COnBlock_CCovered))
)

(:action Pick_2
 :parameters()
 :precondition (and (HandEmpty) (COnTable_CCovered) (BOnBlock) (notfailed))
 :effect (and (BInHand) (COnTable) (HandFull) (not BOnBlock) (not HandEmpty)
         (not COnTable_CCovered) (not BOnBlock) (not HandEmpty)
         (not COnTable_CCovered))
)

(:action Pick_3
 :parameters()
 :precondition (and (HandEmpty) (AOnTable_ACovered) (BOnBlock) (notfailed))
 :effect (and (BInHand) (AOnTable) (HandFull) (not BOnBlock) (not HandEmpty)
         (not AOnTable_ACovered) (not BOnBlock) (not HandEmpty)
         (not AOnTable_ACovered))
)

(:action Pick_4
 :parameters()
 :precondition (and (HandEmpty) (AOnBlock) (BOnBlock_BCovered) (notfailed))
 :effect (and (BOnBlock) (AInHand) (HandFull) (not AOnBlock) (not HandEmpty)
         (not BOnBlock_BCovered) (not AOnBlock) (not HandEmpty)
         (not BOnBlock_BCovered))
)

(:action Pick_5
 :parameters()
 :precondition (and (HandEmpty) (AOnBlock_ACovered) (BOnBlock) (notfailed))
 :effect (and (BInHand) (AOnBlock) (HandFull) (not BOnBlock) (not HandEmpty)
         (not AOnBlock_ACovered) (not BOnBlock) (not HandEmpty)
         (not AOnBlock_ACovered))
)

(:action Pick_6
 :parameters()
 :precondition (and (HandEmpty) (AOnBlock) (BOnTable_BCovered) (notfailed))
 :effect (and (BOnTable) (AInHand) (HandFull) (not AOnBlock) (not HandEmpty)
         (not BOnTable_BCovered) (not AOnBlock) (not HandEmpty)
         (not BOnTable_BCovered))
)

(:action Pick_7
 :parameters()
 :precondition (and (HandEmpty) (BOnTable) (notfailed))
 :effect (and (BInHand) (HandFull) (not BOnTable) (not HandEmpty) (not BOnTable)
         (not HandEmpty))
)

(:action Pick_8
 :parameters()
 :precondition (and (HandEmpty) (COnTable) (notfailed))
 :effect (and (CInHand) (HandFull) (not COnTable) (not HandEmpty) (not COnTable)
         (not HandEmpty))
)

(:action Pick_9
 :parameters()
 :precondition (and (HandEmpty) (AOnTable_ACovered) (COnBlock) (notfailed))
 :effect (and (CInHand) (AOnTable) (HandFull) (not COnBlock) (not HandEmpty)
         (not AOnTable_ACovered) (not COnBlock) (not HandEmpty)
         (not AOnTable_ACovered))
)

(:action Pick_10
 :parameters()
 :precondition (and (HandEmpty) (AOnBlock) (COnTable_CCovered) (notfailed))
 :effect (and (COnTable) (AInHand) (HandFull) (not AOnBlock) (not HandEmpty)
```

```
                    (not COnTable_CCovered) (not AOnBlock) (not HandEmpty)
                    (not COnTable_CCovered))
        )

        (:action Pick_11
         :parameters()
         :precondition (and (HandEmpty) (AOnBlock_ACovered) (COnBlock) (notfailed))
         :effect (and (CInHand) (AOnBlock) (HandFull) (not COnBlock) (not HandEmpty)
                    (not AOnBlock_ACovered) (not COnBlock) (not HandEmpty)
                    (not AOnBlock_ACovered))
        )

        (:action Pick_12
         :parameters()
         :precondition (and (HandEmpty) (COnBlock) (BOnTable_BCovered) (notfailed))
         :effect (and (BOnTable) (CInHand) (HandFull) (not COnBlock) (not HandEmpty)
                    (not BOnTable_BCovered) (not COnBlock) (not HandEmpty)
                    (not BOnTable_BCovered))
        )

        (:action Pick_13
         :parameters()
         :precondition (and (HandEmpty) (COnBlock_CCovered) (BOnBlock) (notfailed))
         :effect (and (BInHand) (COnBlock) (HandFull) (not BOnBlock) (not HandEmpty)
                    (not COnBlock_CCovered) (not BOnBlock) (not HandEmpty)
                    (not COnBlock_CCovered))
        )

        (:action Pick_14
         :parameters()
         :precondition (and (HandEmpty) (COnBlock) (BOnBlock_BCovered) (notfailed))
         :effect (and (BOnBlock) (CInHand) (HandFull) (not COnBlock) (not HandEmpty)
                    (not BOnBlock_BCovered) (not COnBlock) (not HandEmpty)
                    (not BOnBlock_BCovered))
        )

        (:action Put_15
         :parameters()
         :precondition (and (HandFull) (AInHand) (notfailed))
         :effect (and (AOnTable) (HandEmpty) (not AInHand) (not HandFull) (not AInHand)
                    (not HandFull))
        )

        (:action Put_16
         :parameters()
         :precondition (and (HandFull) (BInHand) (notfailed))
         :effect (and (BOnTable) (HandEmpty) (not HandFull) (not BInHand) (not HandFull)
                    (not BInHand))
        )

        (:action Put_17
         :parameters()
         :precondition (and (HandFull) (CInHand) (notfailed))
         :effect (and (COnTable) (HandEmpty) (not HandFull) (not CInHand) (not HandFull)
                    (not CInHand))
        )

        (:action Stack_18
         :parameters()
         :precondition (and (HandFull) (CInHand) (BOnTable) (notfailed))
         :effect (and (BOnTable_BCovered) (COnBlock) (HandEmpty) (not HandFull)
                    (not BOnTable) (not CInHand) (not HandFull) (not BOnTable)
                    (not CInHand))
        )

        (:action Stack_19
         :parameters()
         :precondition (and (HandFull) (COnBlock) (BInHand) (notfailed))
         :effect (and (BOnBlock) (COnBlock_CCovered) (HandEmpty) (not HandFull)
                    (not COnBlock) (not BInHand) (not HandFull) (not COnBlock)
                    (not BInHand))
        )

        (:action Stack_20
         :parameters()
         :precondition (and (HandFull) (AOnBlock) (BInHand) (notfailed))
         :effect (and (BOnBlock) (AOnBlock_ACovered) (HandEmpty) (not HandFull)
                    (not BInHand) (not AOnBlock) (not HandFull) (not BInHand)
                    (not AOnBlock))
        )
```

```
(:action Stack_21
 :parameters()
 :precondition (and (HandFull) (AInHand) (BOnTable) (notfailed))
 :effect (and (BOnTable_BCovered) (AOnBlock) (HandEmpty) (not AInHand)
         (not HandFull) (not BOnTable) (not AInHand) (not HandFull)
         (not BOnTable))
)

(:action Stack_22
 :parameters()
 :precondition (and (HandFull) (CInHand) (BOnBlock) (notfailed))
 :effect (and (BOnBlock_BCovered) (COnBlock) (HandEmpty) (not HandFull)
         (not BOnBlock) (not CInHand) (not HandFull) (not BOnBlock)
         (not CInHand))
)

(:action Stack_23
 :parameters()
 :precondition (and (HandFull) (COnTable) (BInHand) (notfailed))
 :effect (and (BOnBlock) (COnTable_CCovered) (HandEmpty) (not HandFull)
         (not BInHand) (not COnTable) (not HandFull) (not BInHand)
         (not COnTable))
)

(:action Stack_24
 :parameters()
 :precondition (and (HandFull) (AInHand) (COnBlock) (notfailed))
 :effect (and (COnBlock_CCovered) (AOnBlock) (HandEmpty) (not AInHand)
         (not HandFull) (not COnBlock) (not AInHand) (not HandFull)
         (not COnBlock))
)

(:action Stack_25
 :parameters()
 :precondition (and (HandFull) (AOnTable) (CInHand) (notfailed))
 :effect (and (COnBlock) (AOnTable_ACovered) (HandEmpty) (not HandFull)
         (not AOnTable) (not CInHand) (not HandFull) (not AOnTable)
         (not CInHand))
)

(:action Stack_26
 :parameters()
 :precondition (and (HandFull) (AInHand) (COnTable) (notfailed))
 :effect (and (COnTable_CCovered) (AOnBlock) (HandEmpty) (not AInHand)
         (not HandFull) (not COnTable) (not AInHand) (not HandFull)
         (not COnTable))
)

(:action Stack_27
 :parameters()
 :precondition (and (HandFull) (AOnBlock) (CInHand) (notfailed))
 :effect (and (COnBlock) (AOnBlock_ACovered) (HandEmpty) (not HandFull)
         (not AOnBlock) (not CInHand) (not HandFull) (not AOnBlock)
         (not CInHand))
)

(:action Stack_28
 :parameters()
 :precondition (and (HandFull) (AOnTable) (BInHand) (notfailed))
 :effect (and (BOnBlock) (AOnTable_ACovered) (HandEmpty) (not HandFull)
         (not BInHand) (not AOnTable) (not HandFull) (not BInHand)
         (not AOnTable))
)

(:action Stack_29
 :parameters()
 :precondition (and (HandFull) (AInHand) (BOnBlock) (notfailed))
 :effect (and (BOnBlock_BCovered) (AOnBlock) (HandEmpty) (not AInHand)
         (not HandFull) (not BOnBlock) (not AInHand) (not HandFull)
         (not BOnBlock))
)
)
```

## C  LIFTED PDDL DESCRIPTION FOR THE BLOCKS WORLD TASK

In contrast, the lifted representation learned below is far more compact. Below we provide the learned representation for the domain. Again, we manually rename the predicates and types to help with readability.

```
(define (domain BlocksWorld)
        (:requirements :strips :typing)
        (:types hand block)
        (:predicates
                (BlockInHand ?w - block)
                (HandFull ?w - hand)
                (BlockOnBlock ?w - block)
                (BlockOnTable ?w - block)
                (HandEmpty ?w - hand)
                (BlockOnTable_BlockCovered ?w - block)
                (BlockOnBlock_BlockCovered ?w - block)
                (notfailed)
        )
        (:action Pick-partition-0
         :parameters (?w - hand ?x - block)
         :precondition (and (notfailed) (HandEmpty ?w) (BlockOnTable ?x))
         :effect (and (BlockInHand ?x) (HandFull ?w) (not (BlockOnTable ?x))
                (not (HandEmpty ?w)))
        )

        (:action Pick-partition-1
         :parameters (?w - hand ?x - block ?y - block)
         :precondition (and (notfailed) (HandEmpty ?w) (BlockOnBlock ?x)
                        (BlockOnBlock_BlockCovered ?y))
         :effect (and (BlockOnBlock ?y) (BlockInHand ?x) (HandFull ?w)
                (not (BlockOnBlock ?x)) (not (HandEmpty ?w))
                (not (BlockOnBlock_BlockCovered ?y)))
        )

        (:action Pick-partition-10
         :parameters (?w - hand ?x - block ?y - block)
         :precondition (and (notfailed) (HandEmpty ?w) (BlockOnTable_BlockCovered ?x)
                        (BlockOnBlock ?y))
         :effect (and (BlockInHand ?y) (BlockOnTable ?x) (HandFull ?w)
                (not (BlockOnBlock ?y)) (not (HandEmpty ?w))
                (not (BlockOnTable_BlockCovered ?x)))
        )

        (:action Put-partition-0
         :parameters (?w - hand ?x - block)
         :precondition (and (notfailed) (HandFull ?w) (BlockInHand ?x))
         :effect (and (BlockOnTable ?x) (HandEmpty ?w) (not (BlockInHand ?x))
                (not (HandFull ?w)))
        )

        (:action Stack-partition-0
         :parameters (?w - hand ?x - block ?y - block)
         :precondition (and (notfailed) (HandFull ?w) (BlockInHand ?x) (BlockOnTable ?y))
         :effect (and (BlockOnTable_BlockCovered ?y) (BlockOnBlock ?x) (HandEmpty ?w)
                (not (HandFull ?w)) (not (BlockOnTable ?y))
                (not (BlockInHand ?x)))
        )

        (:action Stack-partition-1
         :parameters (?w - hand ?x - block ?y - block)
         :precondition (and (notfailed) (HandFull ?w) (BlockOnBlock ?x) (BlockInHand ?y))
         :effect (and (BlockOnBlock ?y) (BlockOnBlock_BlockCovered ?x) (HandEmpty ?w)
                (not (HandFull ?w)) (not (BlockOnBlock ?x))
                (not (BlockInHand ?y)))
        )

)
```

A task might then be specified as follows:

```
(define (problem stack)
   (:domain BlocksWorld)

   (:objects hand - Hand
          A B C - Block
   )
   (:init (BlockOnTable A)
          (BlockOnTable B)
          (BlockOnTable C)
          (HandEmpty hand)
          (notfailed)
    )
   (:goal (and (BlockOnBlock A)
               (BlockOnBlock_BlockCovered C)
               (BlockOnTable_BlockCovered B)))
)
```

## D  ILLUSTRATION OF LEARNED OPERATORS

In Figure 6, we illustrate one propositional operator for picking block B off block C, while Figure 7 illustrates the lifted version of the same operator for picking any block X off any block Y.

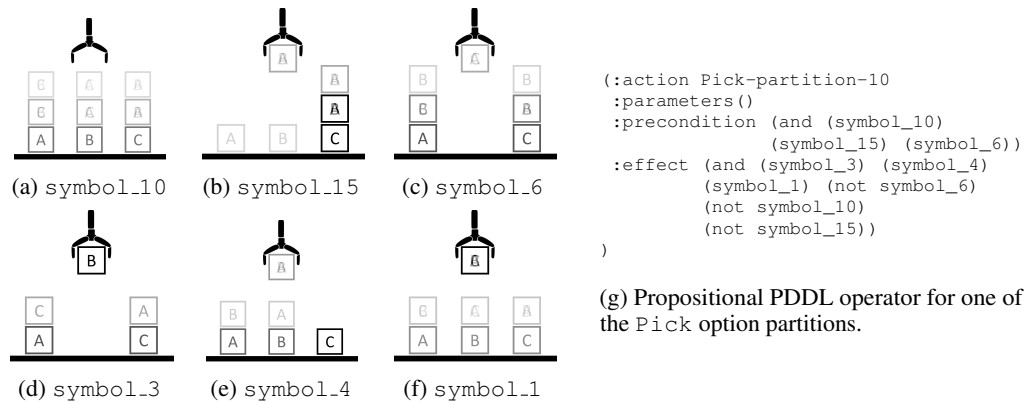

(a) symbol_10   (b) symbol_15   (c) symbol_6

(d) symbol_3   (e) symbol_4   (f) symbol_1

```
(:action Pick-partition-10
 :parameters()
 :precondition (and (symbol_10)
              (symbol_15) (symbol_6))
 :effect (and (symbol_3) (symbol_4)
        (symbol_1) (not symbol_6)
        (not symbol_10)
        (not symbol_15))
)
```

(g) Propositional PDDL operator for one of the Pick option partitions.

Figure 6: The learned propositional operator for a Pick action describing picking B off C. To execute the action, the hand must be empty (symbol_10), C must be on the table and covered by a block (symbol_15), and B must be on top of a block and uncovered (symbol_6). After execution, B is in the hand (symbol_3), C is on the table and clear (symbol_4), and the hand is full (symbol_1). We visualise each propositional symbol by sampling from it, and randomly sampling the remaining independent state variables (since each symbol is a distribution over a subset of state variables). The transparency is due to the averaging over the independent state variables. Note that we must learn one operator for every pair of blocks.

(a) symbol_4    (b) symbol_5    (c) symbol_2

(d) symbol_0    (e) symbol_3    (f) symbol_1

```
(:action Pick-partition-10
 :parameters (?w - type0 ?x - type1
              ?y - type1)
 :precondition (and (symbol_4 ?w)
   (symbol_5 ?x) (symbol_2 ?y))
 :effect (and (symbol_0 ?y) (symbol_3 ?x)
 (symbol_1 ?w) (not (symbol_2 ?y))
 (not (symbol_4 ?w)) (not (symbol_5 ?x)))
)
```

(g) Lifted PDDL operator for a `Pick` action.

Figure 7: The learned lifted operator for a `Pick` action describing picking a block off another. In order to pick up block `Y`, it must be on block `X` which itself is on the table, and the hand must be empty. As a result, the hand is not empty, `Y` is now in the hand, and `X` is on the table and clear. `type0` refers to the "hand" type, while `type1` refers to the "block" type.

## E  MINECRAFT TASK DETAILS

Our Minecraft tasks are procedurally generated, consisting of five rooms with various items positioned throughout. Rooms are connected with either regular doors which can be opened by direct interaction, or puzzle doors which require the agent to pull a lever to open. The world is described by the state of each of the objects (given directly by each object's appearance as a $600 \times 800$ RGB image), the agent's view, and current inventory. Figure 8 illustrates the state of each object in the world at the beginning of one of the tasks.

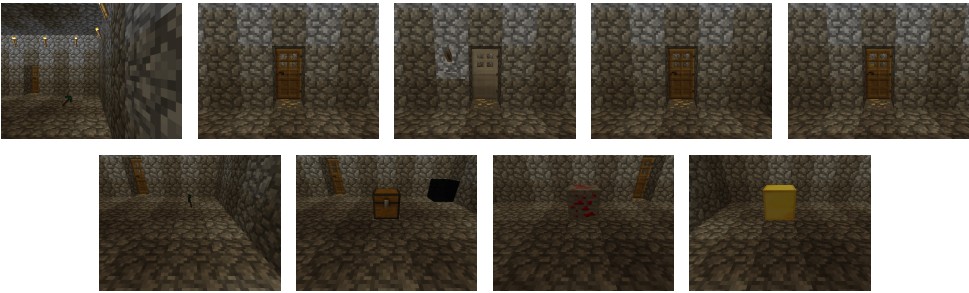

Figure 8: The state of each object in the world at the start of the task. From left to right, the images represent the agent's point of view, the four doors, the pickaxe, the chest, and the redstone and gold blocks. The inventory is not shown here.

The agent is provided with the following high-level skills:

  (i) `WalkToItem`—the agent will approach an item if it is in the same room.

  (ii) `AttackBlock`—the agent will break a block, provided it is near the block and holding the pickaxe.

  (iii) `PickupItem`—the agent will collect the item if it is standing in front of it.

  (iv) `WalkToNorthDoor`—the agent will approach the northern door in the current room.

  (v) `WalkToSouthDoor`—the agent will approach the southern door in the current room.

  (vi) `WalkThroughDoor`—the agent will walk through a door to the next room, provided the door is open.

  (vii) `CraftItem`—the agent will create a new item from ingredients in its inventory, provided it is near the crafting table.

  (viii) `OpenChest`—the agent will open the chest, provided it is standing in front of it and possesses the clock.

  (ix) `ToggleDoor`—the agent will open or close the door directly in front of it.

Execution is stochastic—opening doors occasionally fails, and the navigation skills are noisy in their execution.

## F CRAFTING TASK

The Crafting domain consists of a $12 \times 12$ grid containing the following objects:

  (i) gold, surrounded by water which can only be accessed once a bridge is laid;

 (ii) 4 iron objects;

(iii) 4 grass objects;

(iv) 4 wood objects;

 (v) 4 rock objects; and

(vi) 3 workshops.

Furthermore, a stick can be crafted if the agent has picked up a wood object and is near the second workshop, while a bridge can be crafted if the agent is near the third workshop and possesses wood and iron. Finally, the agent can complete the task by crafting a gold arrow, which requires it to be near the first workshop and holding gold and a stick.

The state of each object is characterised by whether it is on the grid (`present`), has been picked up by the agent (`picked`) or does not yet exist (`non-existent`); the state of the world is given by the state of each object, as well as the agent's egocentric view. The agent is given four skills: `WalkTo`, `Pickup`, `Place` and `Craft`—an object can only be picked up when the agent is adjacent to it, and any particular item can only be crafted if the agent has picked up the correct "ingredients". Furthermore, the gold object is separated by water and can only be accessed once the agent has placed a bridge, which must first be crafted. To solve the task, the agent is required to construct a gold arrow, which involves collecting wood and iron to create a bridge, then collecting the gold, crafting a stick out of wood, and then finally crafting the arrow using the stick and gold.

## G LEARNING A PORTABLE REPRESENTATION FOR MINECRAFT

In this section, we describe the exact details for learning a representation of a Minecraft task. Pseudocode for the approach (independent of the domain) is provided in Appendix I.

In order to learn a high-level representation, we first apply a series of preprocessing steps to reduce the dimensionality of the state space. We downscale images to $160 \times 120$ and then convert the resulting images to greyscale. We apply principal component analysis (Pearson, 1901) to a batch of images collected from the different tasks and keep the top 40 principal components. This allows us to represent each object (except the inventory, which is a one-hot encoded vector of length 5) as a vector of length 40.

**Partitioning**   We collect data from a task by executing options uniformly at random. We record state transition data, as well as which options could be executed at each state. We then partition options using the DBSCAN clustering algorithm (Ester et al., 1996) to cluster the terminating states of each option into separate effects. This approximately preserves the subgoal property as described in Section 2 and previous work (Andersen & Konidaris, 2017; Konidaris et al., 2018; Ames et al., 2018). For each pair of partitioned options, we check whether there is significant overlap in their initiating states (again using DBSCAN). If the initiating states overlap significantly, the partitions are merged to account for probabilistic effects.

**Preconditions**   Next, the agent learns a precondition classifier for each of these approximately partitioned options using an SVM (Cortes & Vapnik, 1995) with Platt scaling (Platt, 1999). We use states initially collected as negative examples, and data from the actual transitions as positive examples. We employ a simple feature selection procedure to determine which objects are relevant to the option's precondition. We first compute the accuracy of the SVM applied to the object the option operates on, performing a grid search to find the best hyperparameters for the SVM using 3-fold cross validation. Then, for every other object in the environment, we compute the SVM's accuracy when that object's features are added to the SVM. Any object that increases the SVM accuracy is kept. Pseudocode for this procedure is outlined in Figure 9.

Having determined the relevant objects, we fit a probabilistic SVM to the relevant objects' data. Note that we learn a single SVM for a given precondition. Thus if the precondition includes two objects, then the SVM will learn a classifier over both objects' features jointly.

```
1:  procedure FEATURESELECTION
2:      Given: affected objects Mask, positive start states p, negative start states n, set of objects M
3:      ▷ Fit a classifier over only objects in the mask
4:      classifier ← FITCLASSIFIER(start, negative, mask)
5:      initScore ← classifier.score
6:      Keep ← ∅
7:      for each object ∈ M \ Mask do
8:          classifier ← FITCLASSIFIER(start, negative, Mask ∪ {object})
9:          newScore ← classifier.score
10:         if newScore > initScore then
11:             ▷ Keep the object if it improves the score
12:             Keep ← Keep ∪ {object}
13:         end if
14:     end for
15:     return Mask ∪ Keep
16: end procedure
```

Figure 9: Pseudocode for a simple feature selection procedure.

**Effects**   A kernel density estimator (KDE) (Rosenblatt, 1956) with Gaussian kernel is used to estimate the effect of each partitioned option. We learn distributions over only the objects affected by the option, learning one KDE for each object. We use a grid search with 3-fold cross validation to find the best bandwidth hyperparameter for each estimator. We fit a single KDE to each object separately, since the state space has already been factored into these objects. Each of these KDEs is an abstract symbol in our propositional PDDL representation.

**Propositional PDDL**   For each partitioned option, we now have a classifier and set of effect distributions (propositions). However, to generate the PDDL, the precondition must be specified in terms of these propositions. We use the same approach as Konidaris et al. (2018) to generate the PDDL: for all combinations of valid effect distributions, we test whether data sampled from their conjunction is evaluated positively by our classifiers. If they are, then that combination of distributions serves as the precondition of the high-level operator. This procedure is described in Figure 10.

**Type Inference**   To determine the type of each object, we first assume that they all belong to their own type. For each object, we compute its effect profile by extracting the effect propositions that occur under each option. Figure 11 illustrates this process.

For each pair of objects, we then determine whether the effect profiles are similar. This task is made easier because certain objects do not undergo effects with certain options. For example, the gold block cannot be toggled, while a door can. Thus it is easy to see that they are not of the same type. To determine whether two distributions are similar, we simply check whether the KL-divergence is less than a certain threshold. Having determined the types, we can simply replace all similar propositions with a predicate parameterised by an object of that type, as described by Figure 12.

```
 1: procedure BUILDPPDDLOPERATOR
 2:     Given: precondition classifier classifier, current effect effect, all effects Effects
 3:        Operators ← ∅
 4:        Symbols ← ∅
 5:        for each candidate ∈ ℘(Effects) do                    ▷ For all possible effect combinations
 6:            samples ← SAMPLE(candidate)                        ▷ Sample from the distributions
 7:            prob ← PREDICT(classifier, sample)                 ▷ Query the classifier with the data
 8:            if prob > 0 then
 9:                if prob = 1 then
10:                    ▷ Construct the new operator with the existing effects
11:                        operator ← {candidate, effect}
12:                else
13:                    ▷ Add a probabilistic failure case
```

$$14: \quad newEffect \leftarrow \begin{cases} \texttt{fail}, \text{with probability } (1 - prob) \\ effect, \text{with probability } prob \end{cases}$$

```
15:                        operator ← {candidate, newEffect}
16:                end if
17:                Operators ← Operators ∪ {operator}
18:                Symbols ← Symbols ∪ {candidate} ∪ {effect}
19:            end if
20:        end for
21:        return Operators, Symbols
22: end procedure
```

Figure 10: Pseudocode for constructing propositional PPDDL operators.

```
 1: procedure COMPUTEEFFECTS
 2:     Given: object i, option o, PPDDL operators Operators
 3:        ▷ Get only the operators that model option o
 4:        Operators ← {operator | ∀operator ∈ Operators, REFERSTO(operator, o)}
 5:        Effects ← ∅
 6:        for each {·, effect} ∈ Operators do
 7:            ▷ Extract the effect propositions that refer to distributions over object i
 8:            OperatorEffect ← {prop | ∀prop ∈ effect, REFERSTO(prop, i)}
 9:            Effects ← Effects ∪ {OperatorEffect}
10:        end for
11:        return Effects
12: end procedure
```

Figure 11: Pseudocode for computing the effect distributions under an option for a given object.

**Problem-Specific Instantiation** Finally, we again use DBSCAN to partition our subgoal options, but this time using problem-specific state variables. Each of these clusters is then added to our representation as a problem-specific proposition. To ground the operators, we add the start and end clusters (problem-specific propositions) to the precondition and effects of the PPDDL operator. We also record the grounded object that appears in the parameter list of each operator, and add a precondition predicate (fluent) to ensure that only *those* particular objects can be modified. Without this final step, the agent would, for example, believe it can open *any* door while standing in front of a door at a particular location. We have thus linked the particular door to a particular location in the domain.

```
 1: procedure MERGE
 2:     Given: objects M, type T, PPDDL operators Operators, propositions Propositions
 3:     ▷ Find the first object matching the type
 4:     archetype ← ∅
 5:     for each object ∈ M do
 6:         if ISTYPE(object, T) then
 7:             archetype ← object
 8:             break
 9:         end if
10:     end for
11:     ▷ Remove propositions with objects of type T that are not the archetype
12:     Removed ← {prop | ∀prop ∈ Propositions, ISTYPE(prop, T),
                        ¬REFERSTO(prop, archetype)}
13:     ▷ Keep operators that do not contain the removed propositions
14:     Operators ← {op | ∀op ∈ Operators, Removed ∩ op = ∅}
15:     return Operators, Propositions \ Removed
16: end procedure
```

Figure 12: Pseudocode for lifting propositions to typed predicates.

## H  LEARNING A REPRESENTATION FOR THE CRAFTING DOMAIN

To learn a representation for the Crafting domain, we use the exact same procedures described in the previous section. However, we use the following hyperparameters:

(i) Partitioning was achieved using DBSCAN with $\epsilon = 0.1$.

(ii) Preconditions were learned using an SVM with $C = 4$.

(iii) The kernel density estimators used for the effects had bandwidth parameter $0.001$.

## I  PSEUDOCODE

Below we present pseudocode describing our approach to building a typed, object-centric PPDDL representation for an arbitrary domain. Some subroutines used in the pseudocode below are outlined in the previous section.

```
 1: procedure LEARNREPRESENTATION
 2:     Given: T state-option transitions D = {(s_i, x_i, o_i, s'_i, x'_i) | 0 ≤ i ≤ T}, set of objects M
 3:     ▷ Partition options into subgoal options
 4:     SubgoalOptions ← ∅
 5:     for each o ∈ O do
 6:         I ← {s | (s, ·, o, ·, ·) ∈ D}                          ▷ Set of initial states for option o
 7:         β ← {s' | (·, ·, o, s', ·) ∈ D}                        ▷ Set of terminating states for option o
 8:         for all K ⊆ I such that Pr(s' | s_i, o) = Pr(s' | s_j, o)∀s_i, s_j ∈ I, s' ∈ β do
 9:             P ← {o, K, {s' | ∀s ∈ K, (s, ·, o, s', ·) ∈ D}}   ▷ Start and end states for a partition
10:             SubgoalOptions ← SubgoalOptions ∪ {P}
11:         end for
12:     end for
13:     ▷ Estimate preconditions and effects
14:     Preconditions, Effects ← ∅
15:     for each {·, start, end} ∈ SubgoalOptions do
16:         mask ← COMPUTEMASK(start, end)                         ▷ List the objects that change state
17:         negative ← S \ start
18:         features ← FEATURESELECTION(mask, start, negative)
19:         classifier ← FITCLASSIFIER(start, negative, features)
20:         Preconditions ← Preconditions ∪ {classifier}
21:         estimator ← FITESTIMATOR(mask, end)                    ▷ Fit over only objects that change
22:         Effects ← Effects ∪ {estimator}
```

```
23:      end for
24:      ▷ Build propositional PPDDL
25:      Operators, Propositions ← ∅
26:      for each precondition, effect ∈ (Preconditions × Effects) do
27:          op, symbols ← BUILDPPDDLOPERATOR(precondition, effect, Effects)
28:          Operators ← Operators ∪ {op}
29:          Propositions ← Propositions ∪ symbols
30:      end for
31:      ▷ Infer object types
32:      EffProfile ← ∅
33:      for each object m do
34:          for each o ∈ 𝒪 do
35:              EffProfile(m, o) ← COMPUTEEFFECTS(m, o, Operators)
36:          end for
37:      end for
38:      Types ← {K | EffProfile(mᵢ, o) ≈ EffProfile(mⱼ, o)∀o ∈ 𝒪, mᵢ, mⱼ ∈ K, K ⊆ ℳ}
39:      ▷ Generate typed PPDDL
40:      TypedOperators, Predicates ← ∅
41:      for each type ∈ Types do
42:          ▷ Replace propositions and operators over objects of same type with lifted versions
43:          ops, predicates ← MERGE(ℳ, type, Operators, Propositions)
44:          TypedOperators ← TypedOperators ∪ ops
45:          Predicates ← Predicates ∪ predicates
46:      end for
47:      ▷ Instantiate typed PPDDL in new task
48:      for each {o, start, end} ∈ SubgoalOptions do
49:          I_𝒳 ← {x | ∀s ∈ start, s' ∈ end, (s, x, o, s', ·) ∈ 𝒟}
50:          β_𝒳 ← {x' | ∀s ∈ start, s' ∈ end, x ∈ I_𝒳, (s, x, o, s', x') ∈ 𝒟}
51:          for all κ ⊆ I_𝒳 such that Pr(x' | xᵢ, o) = Pr(x' | xⱼ, o)∀xᵢ, xⱼ ∈ I_𝒳, x' ∈ β_𝒳 do
52:              λ ← {x' | ∀s ∈ start, s' ∈ end, x ∈ κ, (s, x, o, s', x') ∈ 𝒟}
53:              Predicates ← Predicates ∪ {κ} ∪ {λ}              ▷ Add problem-specific symbols
54:              mask ← COMPUTEMASK(start, end)                  ▷ Computes the affected objects
55:              ▷ Link problem-specific symbols in precondition and effect to the affected objects
56:              TypedOperators ← GROUND(TypedOperators, κ, λ, mask)
57:          end for
58:      end for
59:      return TypedOperators, Predicates
60: end procedure
```

# J    ADDITIONAL RESULTS FOR THE CRAFTING DOMAIN

Based on their effects, each object was grouped into one of the six types below:

| Type | Objects | Remark |
|------|---------|--------|
| type0 | agent | The agent |
| type1 | gold0, iron2, iron3, wood0, wood1, wood2, wood3 | Objects that were picked up and used to craft new items |
| type2 | iron0, iron1, grass0, grass1, grass2, grass3, rock0, rock1, rock2, rock3 | Objects that were picked up but never used in crafting |
| type3 | workshop0, workshop1, workshop2 | Objects that cannot be modified |
| type4 | stick, goldarrow | Objects that can be created but not picked up |
| type5 | bridge | Objects that can be placed |

Table 2: The various objects in the domain along with their learned type.

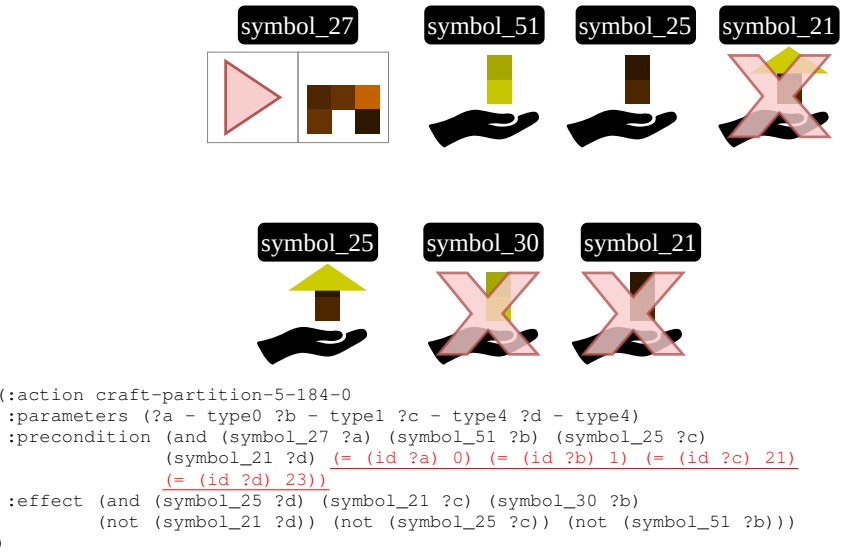

```
(:action craft-partition-5-184-0
 :parameters (?a - type0 ?b - type1 ?c - type4 ?d - type4)
 :precondition (and (symbol_27 ?a) (symbol_51 ?b) (symbol_25 ?c)
                    (symbol_21 ?d) (= (id ?a) 0) (= (id ?b) 1) (= (id ?c) 21)
                    (= (id ?d) 23))
 :effect (and (symbol_25 ?d) (symbol_21 ?c) (symbol_30 ?b)
              (not (symbol_21 ?d)) (not (symbol_25 ?c)) (not (symbol_51 ?b))))
)
```

Figure 13: A learned typed PDDL operator for the Craft skill, which states that, in order to craft a gold arrow, the agent must be in front of a particular workshop (symbol_27), and be in possession of the gold (symbol_51) and stick (symbol_25), but not the gold arrow (symbol_21). As a result, the agent finds itself in possession of the gold arrow (symbol_25), but loses the gold (symbol_30) and stick (symbol_21).

## K  VISUALISATION OF MINECRAFT PLAN

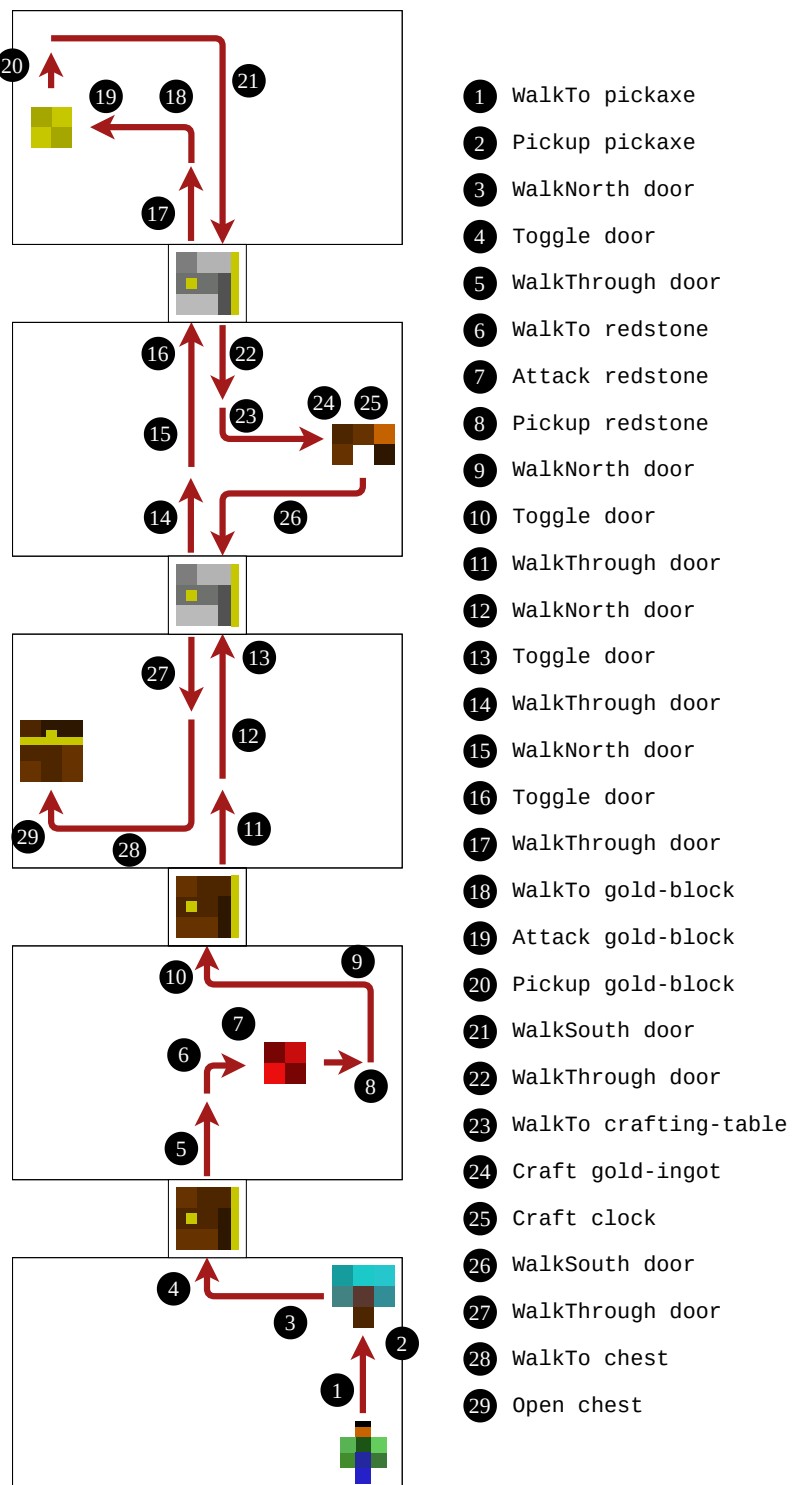

Figure 14: Path traced by the agent executing different options while solving the first task.

## L  VISUALISING OPERATORS FOR MINECRAFT

Here we illustrate some learned operators for the Minecraft tasks. To see all predicates and operators, please see the following URL: `https://sites.google.com/view/mine-pddl`.

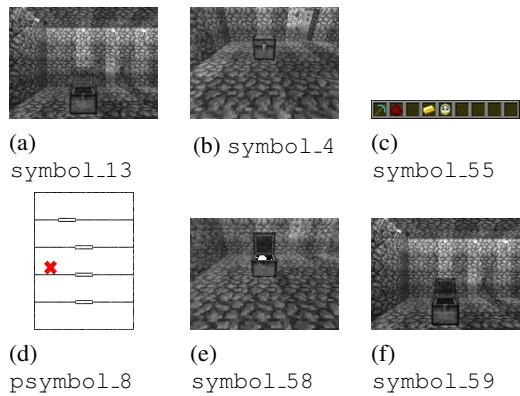

(a)
symbol_13

(b) symbol_4

(c)
symbol_55

(d)
psymbol_8

(e)
symbol_58

(f)
symbol_59

```
(:action Open-Chest-partition-0
:parameters (?w - type0 ?x - type6
              ?y - type9)
:precondition (and (notfailed)
        (symbol_13 ?w) (symbol_4 ?x)
        (symbol_55 ?y) (psymbol_8))
:effect (and (symbol_58 ?x) (symbol_59 ?w)
      (not (symbol_4 ?x))
      (not (symbol_13 ?w)))
)
```

(g) A learned typed PDDL operator for the `Open-Chest` skill. The predicate underlined in red indicates a problem-specific symbol that must be relearned for each new task, while the rest of the operator can be safely transferred.

Figure 15: Our approach learns that, in order to open a chest, the agent must be standing in front of a chest (symbol_13), the chest must be closed (symbol_4), the inventory must contain a clock (symbol_55) and the agent must be standing at a certain location (psymbol_8). The result is that the agent finds itself in front of an open chest (symbol_58) and the chest is open (symbol_59). type0 refers to the "agent" type, type6 the "chest" type and type9 the "inventory" type.

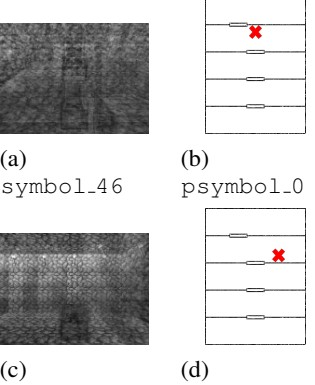

(a)
symbol_46

(b)
psymbol_0

(c)
symbol_11

(d)
psymbol_1

```
(:action Walk-to-partition-0-2a
 :parameters (?w - type0)
 :precondition (and (notfailed) (symbol_46 ?w)
            (psymbol_0))
 :effect (and (symbol_11 ?w) (psymbol_1)
         (not (symbol_46 ?w)) (not (psymbol_0)))
)
```

(e) Typed PDDL operator for a partition of the `Walk-To` option. The predicate underlined in red indicates a problem-specific symbol that must be relearned for each new task, while the rest of the operator can be safely transferred.

Figure 16: Abstract operator that models the agent walking to the crafting table. In order to do so, the agent must be standing in the middle of a room (symbol_46) at a particular location (psymbol_0). As a result, the agent finds itself in front of the crafting table (symbol_1) at a particular location (psymbol_1).

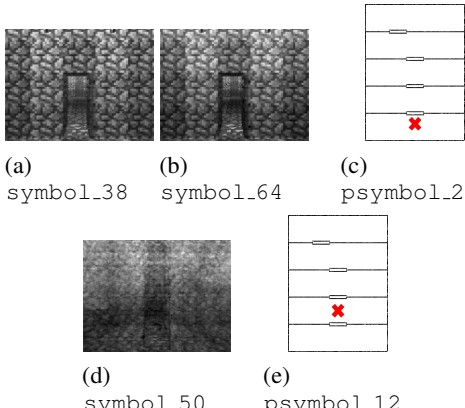

(a)
symbol_38

(b)
symbol_64

(c)
psymbol_24

(d)
symbol_50

(e)
psymbol_12

```
(:action Through-Door-partition-3-207a
 :parameters (?w - type0 ?x - type1)
 :precondition (and (notfailed)
         (symbol_38 ?w) (symbol_64 ?x)
         (= (id ?x) 1) (psymbol_24))
 :effect (and (symbol_50 ?w)
         (not (symbol_38 ?w)) (psymbol_12)
         (not (psymbol_24)))
)
```

(f) Typed PDDL operator for a partition of the Through-Door option. The predicate underlined in red indicates a problem-specific symbol that must be relearned for each new task, while the rest of the operator can be safely transferred.

Figure 17: Abstract operator that models the agent walking through a door. In order to do so, the agent must be standing in front of an open door (symbol_38) at a particular location (psymbol_24), and the door must be open (symbol_64). As a result, the agent finds itself in the middle of a room (symbol_50) at a particular location (psymbol_12).

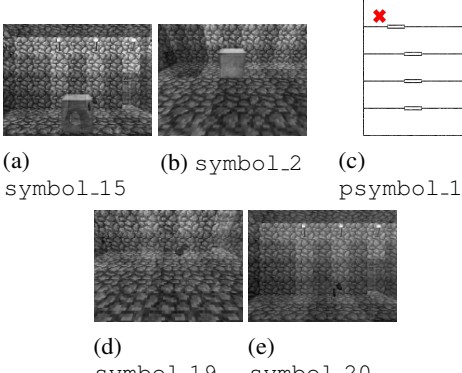

(a)
symbol_15

(b) symbol_2

(c)
psymbol_17

(d)
symbol_19

(e)
symbol_20

```
(:action Attack-partition-0-76a
 :parameters (?w - type0 ?x - type7)
 :precondition (and (notfailed)
      (symbol_15 ?w) (symbol_2 ?x)
      (psymbol_17))
 :effect (and (symbol_19 ?x) (symbol_20 ?w)
         (not (symbol_2 ?x))
         (not (symbol_15 ?w)))
)
```

(f) Typed PDDL operator for a partition of the Attack option. The predicate underlined in red indicates a problem-specific symbol that must be relearned for each new task, while the rest of the operator can be safely transferred.

Figure 18: Abstract operator that models the agent attacking an object. In order to do so, the agent must be standing in front of a gold block (symbol_15) at a particular location (psymbol_17), and the gold block must be whole (symbol_2). As a result, the agent finds itself in front of a disintegrated block (symbol_20), and the gold block is disintegrated (symbol_19).

## M  EXAMPLES OF FAILURE CASES

Below are some examples of errors that occur when constructing our abstract representation. Since there are several phases involving clustering, classification and density estimation, we can expect various learning errors to occur throughout. These errors could have numerous causes, such as insufficient data or suboptimal hyperparameters.

### M.1  PARTITIONING ERRORS

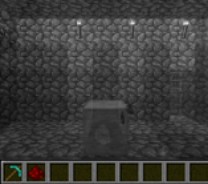

(a) Set of start states for one partition of the `Attack` option.

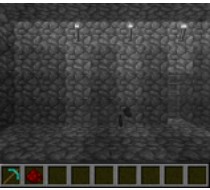

(b) Set of end states for one partition of the `Attack` option.

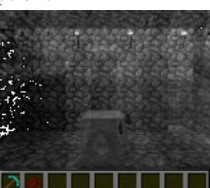

(c) Set of start states for another partition of the `Attack` option.

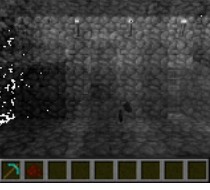

(d) Set of end states for another partition of the `Attack` option.

Figure 19: In the above example, the partitioning procedure has generated two partitioned options for breaking the gold block, where there should only be one. They are functionally equivalent, but because of the strange shadows on the left of the image patch and the subsequent PCA representation, the clustering algorithm has produced one extra partition.

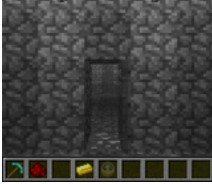

(a) Set of start states for one partition of the `ToggleDoor` option.

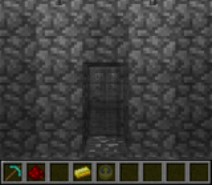

(b) Set of end states for one partition of the `ToggleDoor` option.

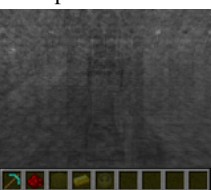

(c) Set of start states for another partition of the `ToggleDoor` option.

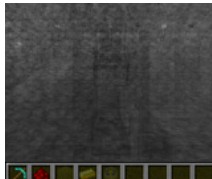

(d) Set of end states for another partition of the `ToggleDoor` option.

Figure 20: In this example, the partitioning has clustered noisy samples into an additional partition of the `ToggleDoor` option. While the top row shows the case where the state of the door changes from open to closed, the bottom row is a relatively useless noisy operator. We will subsequently learn a precondition and effect for this partition, but it likely will not be used by the planner.

## M.2 PRECONDITION/FEATURE SELECTION ERRORS

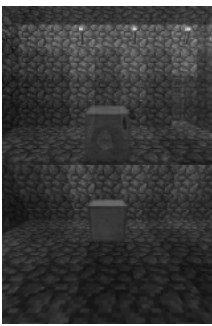

(a) The precondition for attacking the gold block. The top image represents the agent's view (in front of the block), while the bottom image is the state of the block (unbroken).

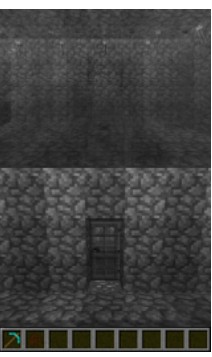

(b) The precondition for walking to a closed door. The top image represents the agent's view (in a room), while the bottom image is the state of the door (closed) and the state of the inventory.

Figure 21: In the left example, the classifier predicts that the gold block can be broken when the agent is in front of it. However, this is not quite correct, since the agent must also have the pickaxe to break the block. In this case, the issue occurs because the data only included states where the agent reached the gold block with the pickaxe. Therefore, the agent did not observe states where it was in front of the block without the pickaxe, and thus concluded that the pickaxe is irrelevant to the precondition. In the right example, the classifier has overfitted to the data and predicts that the agent can only walk to the door when it has the pickaxe.

## M.3 PPDDL CONSTRUCTION ERROR

The quality of the PPDDL operators depends on how accurately the precondition classifiers and effect estimators are learned. Any error in learning can result in imperfect PPDDL operators, as seen below.

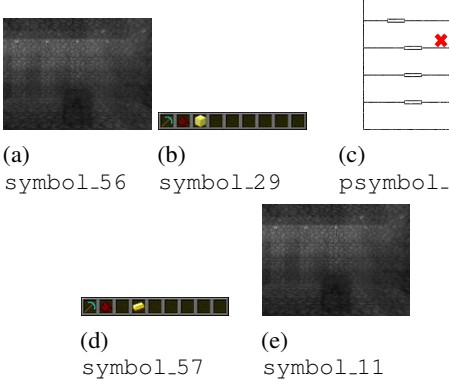

(a)
symbol_56

(b)
symbol_29

(c)
psymbol_1

(d)
symbol_57

(e)
symbol_11

```
(:action Craft-partition-1-240a
 :parameters (?w - type0 ?x - type9)
 :precondition (and (notfailed)
             (symbol_56 ?w) (symbol_29 ?x)
             (psymbol_1))
 :effect (probabilistic 0.21
         (not (notfailed))
                     0.79
         (and (symbol_57 ?x) (symbol_11 ?w)
         (not (symbol_29 ?x))
         (not (symbol_56 ?w))))
)
```

(f) Typed PPDDL operator for a partition of the Craft option.

Figure 22: Abstract operator that models the agent crafting a gold ingot. In order to do so, the agent must be standing in front of the crafting table (symbol_56) at a particular location (psymbol_1), and must have the gold block in its inventory (symbol_29). As a result, the agent finds itself in front of the crafting table (symbol_11) with a gold ingot in its inventory (symbol_57). This option is deterministic—however, due to estimation errors, the PPDDL operator predicts that it will only succeed with probability 0.79.

## M.4 TYPE INFERENCE ERROR

We observe that occasionally the procedure will not discover the correct types. In the example below, instead of discovering a single type for all four doors, our approach predicts that one door is different from the others and is placed in its own class

| Type | Name | Object(s) |
|------|------|-----------|
| 0 | Agent | 0 |
| 1 | Pickaxe | 1 |
| 2 | Door1 | 2, 3, 4 |
| 3 | Door2 | 5 |
| 4 | Redstone Block | 6 |
| 5 | Gold Block | 7 |
| 6 | Chest | 8 |
| 7 | Inventory | 9 |

Table 3: A grouping of objects into types. Note that one of the doors is allocated its own type.

