# OpenReview forum: "Autonomous Learning of Object-Centric Abstractions for High-Level Planning"
_ICLR.cc/2022/Conference — ICLR 2022 Poster_

### Official Review · Reviewer_nhv5 · 2021-10-29

**Correctness:** 4
**Technical Novelty And Significance:** 3
**Empirical Novelty And Significance:** 3
**Recommendation:** 8
**Confidence:** 3

**Main Review:**

The focus of this paper is on learning representations of an environment that allow for planning, which is an important problem. The main driver for the presented improvements is the assumption that the world can be organized according to objects that are relevant for many different tasks. Recently, there has been an increasing interest in incorporating such real-world priors in representation learning methods, which makes the overall focus highly relevant. The paper is also well written and the ideas are presented clearly. Further, the paper does a good job at crediting prior work wherever appropriate.

The main improvements in this paper compared to prior work of Konidaris et al. (2018) are

1. A method for estimating object types by evaluating their effect distributions when applying (subgoal) options.

2. A method for integrating task-specific information in the learned representations, i.e. grounding them for particular problem instantiations.

I think (1) is quite clever since it allows for a grouping into object types that focuses on their functionality, which transcendent a grouping based on other features such as appearance. On the other hand, it is not quite clear to me why a grouping based on effect distributions alone is preferred over one that also considers the pre-condition? The assumption that an option only affects the object it acts upon is quite strong and in my view somewhat unrealistic, yet it wasn't quite clear to me how this relates to this choice either.

Regarding (2), I find that the current presentation falls somewhat short. For example, it is unclear to me how it is determined what objects/operators require grounding (is this prior knowledge that is assumed given. Eg. for the workbench or door in the experiments). It is also not clear to me how the clustering proceeds for X. In particular, is X still partitioned according to objects similar to S? And how is the result from clustering X connected to clustering S, i.e. how are the clusters mapped onto each other and how is it ensured that a similar split is obtained? I am particularly interested in understanding how much domain knowledge is injected in (2). It may very well be that I misunderstood something here, so further clarification is much appreciated.

The approach is evaluated on three different environments. On Blocks World it is demonstrated how the 'correct' environment representation is learned and the benefits from merging into object types is clearly demonstrated.

On the Craft environment from Andreas et al. (2017) it is shown how the approach scales to situations where there are many different objects, and further how grounding can be achieved for operators that require identities for a specific problem instantiation. Here the identities are derived from the option masks, and it is shown how a grounded operator is learned. However, similar to above it is unclear to me what are the steps needed to achieve this result, i.e. how is the need for having identities identified (or is it assumed given?), how is it determined which objects require ids (or is it assumed given?), and how is the id connected to the object. What I liked about this experiment is how not the 'ground truth' types are recovered but rather a suboptimal grouping based on the data collected so far. It also suggests that it may be worthwhile to relearn the object types after having interacted with the environment for a number of steps, which is something that could be commented on.

On the MineCraft environment from Johnson et al. (2016) it is shown how this approach can be combined when learning feature representations of objects from their pixel representation rather than using pre-specified features. Here each object's appearance is given by a high-dimensional image although it is still assumed that objects are pre-segmented. Here, the learned operators must be grounded to consider a particular door (and I have similar questions as above). By considering 5 procedurally generated tasks it is also demonstrated how object types allow for transfer compared to prior work of Konidaris et al. (2018), which clearly demonstrates the benefits of this approach.

As a final comment, I would have liked to see some more discussion of the limitations of this approach. The appendix includes several failure cases, which are interesting, but the framework presented here makes several strong assumptions that are worth commenting on: (1) the frame assumption, which appears crucial for learning effect distributions, (2) that options only affect the object they act upon, (3) that the world comes pre-segmented according to objects. When also inferring the objects themselves (i.e. the segregation process) the merging into object types may affect one another, since by adopting a different notion of an object their effect distribution is likely to change.


**Summary Of The Paper:**

This paper proposes a method for learning an object-centric symbolic representation of an environment that allows for planning. It extends the framework introduced in Konidaris et al. (2018), which learns a symbolic representation of an environment that can be expressed using a PDDL for planning. Importantly, the method presented in Konidaris et al. (2018) does not make assumptions about structure in the environment and how different states may relate to one another. As a consequence, the learned representations are highly tasks-specific and transfer/generalization to new tasks and across states is not possible.

In this paper it is assumed that the world consists of objects and that similar objects are common across tasks. This enables object-centric representations (and the associated model) to be reused across tasks, and organizing objects according to 'object types' that behave similarly when acted upon. This is expected to improve learning efficiency and generalization/transfer to new tasks.

The method presented here incorporates techniques for merging objects into object types, and integrating problem-specific information to ground these representations into specific tasks. The working of these techniques is validated experimentally, and it is shown how greater sample efficiency can be obtained when generalizing to new tasks.

**Summary Of The Review:**

This is a well-written paper that introduces several techniques into an existing framework for learning symbolic representations for planning. The proposed contributions are interesting and the improvement over prior work is clearly demonstrated.

While the final approach itself is still limited, in the sense that it relies on several strong assumptions, it advances what symbolic approaches are currently capable of. Further, by highlighting the trade-off between creating object types and problem-specific grounding it contributes an interesting problem that may not yet be on everyone's radar.

While I would like to see this paper accepted I also think it can still be improved. The presented approach is quite complex, and although the authors do a good job overall, the presentation for the grounding is lacking and it is not made sufficiently clear how this is achieved and what knowledge is assumed to arrive at the presented results. I also suggest that the authors consider creating a figure to accompany figure 1 that specifically visualizes step 1 (and step 5). In order to make the paper accessible to a wider audience, an introduction to PPDDL notation would be helpful to include.

---

> ### Author Response · Authors · 2021-11-17
> **Response to Reviewer nhv5 (Part 1)**
>
> Thank you for your time and considered report.  As a general comment, we are grateful for the reviewer pointing out gaps in the presentation or where things are not clear. Unfortunately the ICLR page limits and single column format are extremely limiting in this regard (especially with a lot of non-standard background material to cover), but we will update and improve the paper to clarify the various issues, and provide more details in the appendix should it not fit. To address the questions and comments here:
>
> > On the other hand, it is not quite clear to me why a grouping based on effect distributions alone is preferred over one that also considers the pre-condition? The assumption that an option only affects the object it acts upon is quite strong and in my view somewhat unrealistic, yet it wasn't quite clear to me how this relates to this choice either.
>
> You are correct in that considering the preconditions would be stronger, but it could also result in more redundant types, and we find that effects alone are sufficient for the most part (although there could definitely be adversarial cases constructed). Reviewer UVNz actually raises a good example of why using preconditions could be a less attractive option. If we have an apple that can be picked off different surfaces, then we will have multiple PDDL operators. For example, one for picking off a table, one for picking from a tree, etc. So now instead of having just a single “apple” object, we will end up with “appleOffTable”,  “appleOffTree”, etc objects. While this is not incorrect, it is a bit superfluous, and may hurt the generality of the representations.
>
> Although we make the assumption about actions affecting only that object for the purpose of computing types, our experiments actually violate this. For example, when we pickup an object in Minecraft, the agent’s view changes, the object’s state changes (it disappears) and the inventory changes (the object is added to it). This is still okay, because the effect of the action on the object it interacts with is that the object disappears, and this is what is used to determine the type. Perhaps a better way of stating this assumption is that when we interact with an object, at least that object’s state changes (but there may be others). We will include this point in an updated revision.
>
> > Regarding questions about object groundings and IDs:
>
> In theory, we are using data already collected to do the grounding, so if we decide to ground all the objects and add in their IDS to every operator, then that is okay from a sample efficiency point of view. In practice, we do the grounding only for objects that are “ambiguous”. For example, if objects X and Y are both estimated to be of type “door”, then we would apply the grounding procedure to both since it likely matters whether it is door1 or door2 that we are walking through. But if the object is the only one of its type, then we don’t need to bother since it cannot be confused with a different object of the same type. In either case, the agent determines this by itself.
>
> The ID of the object for a given task is its location in the state representation. Our state representation is the vector $\langle f_0, f_1, \ldots \rangle$. Let’s say the agent executed an open-door option. Then, based on the change in state vector, we can determine which of the elements $f_i$ have changed (by the frame assumption, everything else remains the same) and so the agent then knows that the object it interacted with has ID $i$.
>
> > In particular, is X still partitioned according to objects similar to S? And how is the result from clustering X connected to clustering S, i.e. how are the clusters mapped onto each other and how is it ensured that a similar split is obtained? I am particularly interested in understanding how much domain knowledge is injected in (2)
>
> Yes, it is almost an identical procedure (although in practice we use different hyperparameters for clustering). Essentially we start out by collecting transition data of the form $(s, x), option, (s’, x’)$ where $s$ and $x$ are the states (and the primes are next states) in $S$ and $X$ respectively. So from the data itself, we have the association between individual states. Then we initially cluster based on $S$. For each cluster in $S$, we hold onto the $X$ data that was in the original dataset and was associated with every $s$ in the given cluster. So the association comes through the way the data is collected. Then, for each cluster in $S$, we can take all the $X$ data, and cluster just that, meaning that for each $S$ cluster, we now have multiple $X$ clusters associated with it. As a concrete example, $S$ clusters may consist of one cluster for a door being open, and one for a door being closed. Each of these will have the $xy$ positions whenever the agent was in front of an open or closed door respectively. Then clustering on $xy$ data produces multiple clusters tied to “door open” and “door closed”.

---

> > ### Author Response · Authors · 2021-11-17
> > **Response to Reviewer nhv5 (Part 2)**
> >
> > > What I liked about this experiment is how not the 'ground truth' types are recovered but rather a suboptimal grouping based on the data collected so far. It also suggests that it may be worthwhile to relearn the object types after having interacted with the environment for a number of steps, which is something that could be commented on.
> >
> > This is a great point and something that we can add a discussion on. One thing it also suggests as a general principle is that there is no one true objective answer for what type an object should be - it is induced by the agent. For example, as humans we would consider a door and wall to be different types of objects, but that’s only because we can open a door. But if we were a navigating robot like a roomba, and we did not have the ability to open or close with a door, then doors and walls are identical (since they are just obstacles that cannot be passed through). This is of course very related to the notion of object affordances. Collecting more data would definitely be beneficial, not just for refining the types of objects, but also for constructing more robust preconditions and effects. Ideally, we would have an online framework that could accept transition data on the fly and construct and update representations in the background, but this is a significant engineering challenge and we leave it to future work.
> >
> >
> > > I would have liked to see some more discussion of the limitations of this approach. The appendix includes several failure cases, which are interesting, but the framework presented here makes several strong assumptions that are worth commenting on...
> >
> > We have tried to indicate in the appendix that not all is perfect and that there are failure cases and imperfections (as there will be with learning methods), but these are good points and we can certainly add discussions around them.
> >
> > The object-segmentation is obviously the biggest one - our claim in the paper is essentially that “if we can do this object segmentation, then here is how it can be leveraged”. Very recently, approaches like SORNet [1] have shown that this can be done, and it would be natural to include such an approach in the pipeline.  On the frame assumption, this is a standard one that motivates approaches such as STRIPS. Of course, the real world is not like this, but interestingly, because we support probabilistic PDDL, we would find that if this assumption were violated, we would still be able to function, but our effects would appear probabilistic e.g. with 0.5 probability, the door opens and object X in the background moves, and with probability 0.5, the door opens and object Y moves.
> >
> >
> > > I also suggest that the authors consider creating a figure to accompany figure 1 that specifically visualizes step 1 (and step 5). In order to make the paper accessible to a wider audience, an introduction to PPDDL notation would be helpful to include.
> >
> > Thank you for the suggestions - these are great ideas! We will definitely include both, and will try to fit them in the main text. Failing that, the image will be a great addition, and the PPDDL notation may need to be moved to the appendix.
> >
> > [1] Yuan, Wentao, et al. "SORNet: Spatial Object-Centric Representations for Sequential Manipulation." arXiv preprint arXiv:2109.03891 (2021).

---

> > > ### Comment · Reviewer_nhv5 · 2021-11-30
> > > **Reply**
> > >
> > > Thank you for your detailed reply and the additional clarification. I have no further concerns and look forward to the revised version that incorporates these changes!

---

### Official Review · Reviewer_PYjf · 2021-10-31

**Correctness:** 3
**Technical Novelty And Significance:** 3
**Empirical Novelty And Significance:** 3
**Recommendation:** 6
**Confidence:** 3

**Main Review:**

This paper is overall well-written, clearly presented and generally of high quality. The problem of learning sample-efficient abstractions for long-term planning is an important and difficult problem, and taking an object-centric approach to address this problem is very timely and of high significance. The proposed method is largely based on a framework introduced in prior work (Konidaris et al., 2018), but introduces sufficient novelty in terms of integrating object-centricity in an elegant way into this framework. The experimental evaluation is carried out on sufficiently complex environments to demonstrate that the method can solve tasks that are not trivial.

My main concern with this paper is with regards to its sample complexity claims and limited experimental comparison. The paper claims (in the abstract) that the proposed method results in being able to obtain a successful agent for long-term planning „with considerably fewer environment interactions“. Given this central statement, I would expect some form of quantitative evaluation against a baseline that demonstrates this strong reduction in environment interactions (which unfortunately is not provided). Potentially it is trivial to see that this method uses fewer environment interactions than prior works (for example the works by Kaiser et al. (2020) and Hafner et al. (2021), as discussed in the paper) — this would not be a fair comparison, however, as the presented method assumes that environment observations are pre-factored and pre-processed into object components (e.g. provided as abstract per-object feature vector or as cropped images around individual objects). Extracting object information in this way without direct supervision is a highly non-trivial task, which might partially explain the sample inefficiency of earlier methods (Kaiser et al., 2020 and Hafner et al., 2021). This present paper sidesteps this issue by providing ground truth information. I would recommend either 1) significantly reducing the prominence of the sample efficiency claims in the paper or 2) providing a fair experimental comparison against a baseline that has access to the same factored, pre-processed observations and abstract options (instead of low-level actions) as the presented technique.


**Summary Of The Paper:**

This paper introduces a method for learning symbolic, object-centric abstractions from object-factored environment observations for long-term planning tasks. It extends the symbolic representation learning framework by Konidaris et al. (2018) by factoring the state into objects, learning object type abstractions, and “lifting” the model to operate on these object-centric, typed abstractions. The method is successfully demonstrated on three environments of complexity ranging from simple per-object discrete feature vectors to a Minecraft long-horizon planning task from (object-factored) pixel observations.

**Summary Of The Review:**

Despite the above-mentioned limitations, I think that this paper carries sufficient novelty and is an interesting, significant extension of prior work in this area, which opens up interesting follow-up questions and opportunities for future work. Overall, I recommend “weak accept”.

---

> ### Author Response · Authors · 2021-11-17
> **Response to Reviewer PYjf**
>
> Thank you for the time you spent reviewing the paper! Your point regarding experiments is well taken. We are quite constrained by the computing hardware we have at our disposal, and the fact that the Malmo platform is *extremely* slow to interact with. The Minecraft results presented here took weeks to produce because of this. One point to make is that the “No transfer” bar graph is produced using Kondaris et al (2018), and while it does not leverage the factorised state space (the state space is “flattened”), it does use the options. Nonetheless, we agree that a more thorough experiment would be better.
>
> We did attempt to use the data collected already and apply BCQ, which is a batch RL method [1]. In this instance, the state space was the flattened vector of PCA representations, and the actions were the options. Unfortunately, we haven’t produced usable results as the agent fails to learn anything. We suspect this is as a result of the very sparse reward function (+1 at the goal, -1 everywhere else).
>
> In the meantime, we will update the paper to soften the language, and report back should we have new results/baselines to include to strengthen the paper.
>
> [1] Fujimoto, Scott, David Meger, and Doina Precup. "Off-policy deep reinforcement learning without exploration." International Conference on Machine Learning. PMLR, 2019.

---

> > ### Comment · Reviewer_PYjf · 2021-11-17
> > **Re: Response to Reviewer PYjf**
> >
> > Thanks a lot for your detailed response. I very much appreciate the explanation around your experiments in the Minecraft environment and I am looking forward to reading the revised version of the paper.
> >
> > As I mentioned in my review, while I think that additional experimental evidence would certainly strengthen the point around sample efficiency in the paper, I think the paper has sufficient novelty to meet the bar for acceptance even if this point is de-emphasized.

---

### Official Review · Reviewer_UVNz · 2021-11-04

**Correctness:** 3
**Technical Novelty And Significance:** 2
**Empirical Novelty And Significance:** 2
**Recommendation:** 5
**Confidence:** 4

**Main Review:**

The main contribution of this paper is that the author devises a learning framework to abstract object types, pre and post-conditions by grouping object/symbol instances that are option-equivalent or effect-equivalent. Although the proposed method, to some extent, alleviates human efforts to define the planning domain manually, it has two main drawbacks that cannot solve the fundamental problem of the human-defined planning domain.

1. The author claims the learned object-centric abstraction can be transferred to a new task with the same object type. However, it makes no sense if we cannot define how same is w.r.t. the same object type in the new task. For example, we learned the abstraction from picking an apple on the table, and now, given a new task, we want to pick an apple on a tree. These two apples are the same in semantics, but they are different in the planning domain as they have different preconditions for the `pick` action. Such object-centric abstractions are still more of task-specific representation, even for the same objects in semantics. As such, very likely, they have different constraints (preconditions) that limit the actions across different tasks.

2. As mentioned in the paper, we could use Problem-Specific Instantiation tricks to solve the problem of the same-type object but have different preconditions in a different scenario, i.e., adding some task-specific constraints for the object types in different scenarios. However, such a process is exactly the same as how we manually maneuver the planning domain to adapt to a different scenario, and it does not solve the fundamental limitation of the human-defined planning domain: Learned symbolic predicates of specific object types are not general enough to across different task. However, the proposed object-centric abstraction has the same issue. It only groups the similar conditions it observed, but does not actually abstract the preconditions of why an action could be performed. Like the aforementioned picking apple example, if we place the apple on $N$ different heights, the proposed method will lead to $N$ different apple object types as they have N different preconditions. If we place the apple even higher--a condition out of the $N$ previously seen cases, the proposed method still cannot solve this problem as it does not abstract the `reachable` concept for the picking action.

**Summary Of The Paper:**

This paper proposes a method to automatically learn a PDDL-like abstractions for high-level problem solving based on the different states (either symbolic representation, or high-dimensional embeddings) of the objects in the environment. Such abstractions could support a PDDL solver to find a solution for the task where the abstractions were learned, and the authors claim such abstractions could be shared among different tasks as long as object types remain the same.

**Summary Of The Review:**

The authors proposed a learning framework to learn object-centric abstractions for high-level planning, yet it does not solve the fundamental limitation of the manually designed planning domain, and it is more of an autonomous labeling tool that partially alleviates human efforts.

---

> ### Author Response · Authors · 2021-11-17
> **Response to Reviewer UVNz**
>
> Thank you for your time and effort spent reviewing the paper. You raise an excellent point about learning these planning-centric representations. However, we argue that this is not a failing of our method in particular, but is rather a problem inherent when an agent is required to learn from limited data.
>
> Putting aside the notion of objects and types for a moment, if we attempt to learn PDDL operators (e.g. using the approach of Konidaris et al (2018)) for picking an apple in the situation described, then the following would occur. In task 1, the agent would collect data of it picking up an apple off the table, which would then be used to train a precondition classifier. Because the only positive examples of picking up an apple is when it is on the table, the classifier will naturally only return true in these situations. In the second task, when the apple is on the tree, the classifier would almost certainly be unable to generalise (having never seen states where an apple is on a tree) and therefore return false. This could be rectified by collecting more data in the second task and relearning, but absent this, the operator learned in the previous task would not be usable in the second task. Again, though, this is because of a lack of data and the inability of classifiers to generalise far outside their experience.
>
> Coming back to our work in particular, we categorise objects into types based on effects. Assuming that the effect of the “pick” action is that the apple is “in hand”, then whether apples are picked from a table or tree (or different heights), the final effect of that action is that it is “in hand” and so the objects would all be grouped into a *single* type (perhaps called “pickable”). However, as you say, this does not help when an agent observes a situation that has not been described by previously learned preconditions. If the agent has observed enough data (apples picked off of tables, trees, different heights), then that data would be used to train a classifier that will return true in all of those cases (and perhaps be slightly more robust and general).  However, if this is not the case, then it all comes down to whether the classifier is able to generalise well enough given limited data. Perhaps the classifier is able to generalise and determine that the apple can always be picked when there is a red circle-like object in the agent’s field of view.
>
> In any case, the failure here is not with our specific method or definition of types, but rather about learning from limited samples. Of course, if you have a human available, and they're skilled enough to define the domain, they should! But that definitionally limits agent autonomy --- truly autonomous agents will be able to learn the domain description themselves, without human help. Such agents will necessarily have to generalise from the limited samples they have already seen, which is a fundamental challenge in machine learning as a whole.

---

### Decision · Program_Chairs · 2022-01-20

**Decision:**

Accept (Poster)

**Comment:**

This paper extends the symbolic representation learning work of Konidaris et al. (2018) to be object-centric, and generalize in this respect. All the reviewers agreed that this is an interesting problem, and that the approach is novel.
Two reviewers gave positive evaluations (6,8), and one reviewer gave a mildly negative review (5), where the main critique is that the method still requires some human effort in designing the planning domain.
While completely alleviating human efforts is definitely a good goal to pursue, I believe that it's too high a bar to ask for in the setting of limited data, and I believe that there are many real world problems where requiring some human effort is not too limiting.
Therefore I recommend acceptance.
Please take all reviewer comments into account when preparing the final version.